

# Antarctic sub-shelf melt rates via PICO

Ronja Reese[1,2], Torsten Albrecht[1], Matthias Mengel[1], Xylar Asay-Davis[1], and Ricarda Winkelmann[1,2]

[1] Potsdam Institute for Climate Impact Research (PIK), Member of the Leibniz Association, P.O. Box 60 12 03, D-14412 Potsdam, Germany

[2] University of Potsdam, Institute of Physics and Astronomy, Karl-Liebknecht-Str. 24-25, 14476 Potsdam, Germany

*Correspondence to:* Ricarda Winkelmann (ricarda.winkelmann@pik-potsdam.de)

**Abstract.** Ocean-induced melting below ice shelves is one of the dominant drivers for mass loss from the Antarctic Ice Sheet at present. An appropriate representation of sub-shelf melt rates is therefore essential for model simulations of marine-based ice sheet evolution. Continental-scale ice sheet models often rely on simple melt-parameterizations, in particular for long-term simulations, when fully coupled ice-ocean interaction becomes computationally too expensive. Such parameterizations can
account for the influence of the local depth of the ice-shelf draft or its slope on melting. However, they do not capture the effect of ocean circulation underneath the ice-shelf. Here we present the Potsdam Ice-shelf Cavity mOdel (PICO), which simulates the vertical overturning circulation in ice-shelf cavities and thus enables the computation of sub-shelf melt rates consistent with this circulation. PICO is based on an ocean box model that coarsely resolves ice shelf cavities and uses a boundary layer melt formulation. We implement it as a module of the Parallel Ice Sheet Model (PISM) and evaluate its performance under
present-day conditions of the Southern Ocean. The two-dimensional melt rate fields provided by the model reproduce the typical pattern of comparably high melting near the grounding line and lower melting or refreezing towards the calving front. PICO captures the wide range of melt rates observed for Antarctic ice shelves, with an average of about $0.1\,\mathrm{m\,a^{-1}}$ for cold sub-shelf cavities, for example underneath Ross or Ronne ice shelves, to $12\,\mathrm{m\,a^{-1}}$ for warm cavities such as in the Amundsen Sea region. This makes PICO a computationally-feasible and more physical alternative to melt parameterizations purely based
on ice draft geometry.

## 1   Introduction

Dynamic ice discharge across the grounding lines into floating ice shelves is the main mass loss process of the Antarctic Ice Sheet. Surrounding most of Antarctica's coastlines, the ice shelves themselves lose mass by ocean-induced melting from below or calving of icebergs (Depoorter et al., 2013; Liu et al., 2015). Observations show that many Antarctic ice shelves are thinning
at present, driven by enhanced sub-shelf melting (Pritchard et al., 2012; Paolo et al., 2015). Thinning reduces the ice shelves' buttressing potential, *i.e.*, the restraining force at the grounding line provided by the ice shelves (Thomas, 1979; Dupont and Alley, 2005; Gudmundsson et al., 2012), and can thereby accelerate upstream glacier flow. The observed acceleration of tributary glaciers is seen as the major contributor to the current mass loss in the West Antarctic Ice Sheet (Pritchard et al., 2012). In particular, the recent dynamic ice loss in the Amundsen Sea sector (MacGregor et al., 2012; Mouginot et al., 2014)
is associated with high melt rates that result from inflow of relatively warm circumpolar deep water (CDW) in the ice shelf



cavities (Holland et al., 2008a; Jacobs et al., 2011; Pritchard et al., 2012; Schmidtko et al., 2014; Hellmer et al., 2017). Also in East Antarctica, particularly at Totten glacier, as well as along the Southern Antarctic Peninsula, glacier thinning seems to be linked to CDW reaching the deep grounding lines (Greenbaum et al., 2015; Wouters et al., 2015). An appropriate representation of melt rates at the ice-ocean interface is hence crucial for simulating the dynamics of the Antarctic Ice Sheet. Melting in ice-

shelf cavities can occur in different modes that depend on the ocean properties in the proximity of the ice shelf, the topography of the ocean bed and the ice-shelf subsurface (Jacobs et al., 1992). Antarctica's ice shelf cavities can be classified into "cold" and "warm" with typical mean melt rates ranging from $\mathcal{O}(0.1 - 1.0)\,\mathrm{m\,a^{-1}}$ in "cold" cavities as for the Filchner-Ronne Ice Shelf and $\mathcal{O}(10)\,\mathrm{m\,a^{-1}}$ in "warm" cavities like the one adjacent to Pine Island Glacier (Joughin et al., 2012). For the "cold" cavities of the large Ross, Filchner-Ronne and Amery ice shelves, freezing to the shelf base is observed in the shallower areas

near the center of the ice shelf and towards the calving front (Rignot et al., 2013; Moholdt et al., 2014).

Since the stability of the ice sheet is strongly linked to the dynamics of the buttressing ice shelves, it is essential to adequately represent their mass balance. A number of parameterizations with different levels of complexity have been developed to capture the effect of sub-shelf melting. Simplistic parameterizations that depend on the local ocean and ice-shelf properties have been applied in long-term and large-scale ice sheet simulations (Joughin et al., 2014; Martin et al., 2011; Pollard and DeConto,

2012; Favier et al., 2014). These parameterizations make melt rates piece-wise linear functions of the depth of the ice-shelf draft (Beckmann and Goosse, 2003) or of the slope of the ice-shelf base (Little et al., 2012). Other models describe the evolution of melt-water plumes forming at the ice-shelf base (Jenkins, 1991). Plumes evolve depending on the ice-shelf draft and slope, sub-glacial discharge and entrainment of ambient ocean water. This approach has been applied to models with characteristic conditions for Antarctic ice shelves (Holland et al., 2007; Payne et al., 2007) and for Greenland outlet glaciers and fjord

systems (Jenkins, 2011; Carroll et al., 2015). Interactively coupled ice-ocean models that resolve both the ice flow and the water circulation below ice shelves are now becoming available (Goldberg et al., 2012; Thoma et al., 2015; Seroussi et al., 2017; De Rydt and Gudmundsson, 2016). There is a community effort to better understand effects of ice-ocean interaction in such coupled models (Asay-Davis et al., 2016). However, as ocean models have many more degrees of freedom than ice sheet models and require for much shorter time steps, coupled simulations are currently limited to short time scales (on the order of

decades to centuries).

Here, we present the Potsdam Ice-shelf Cavity mOdel (PICO), which provides sub-shelf melt rates in a computationally efficient manner and resolves the basic vertical overturning circulation in ice shelf cavities driven by the ice pump (Lewis and Perkin, 1986). It is based on the earlier work of Olbers and Hellmer (2010) and is implemented as a module in the Parallel Ice Sheet Model (PISM: Bueler and Brown, 2009; Winkelmann et al., 2011)[1]. Ocean temperature and salinity at the depth of

the bathymetry in the continental shelf region serve as input data. PICO allows for long-term simulations (on centennial to millennial time scales) and for large ensembles of simulations which makes it applicable, for example, in paleo-climate studies or as a coupling module between ice-sheet and Earth System models.

In this paper, we give a brief overview of the cavity circulation and melt physics and describe the ocean box geometry in PICO and implementation in PISM in Sect. 2. In Section 3, we derive a valid parameter range for present-day Antarctica

---

[1]http:\\www.pism_docs.org



**Table 1.** PICO parameters and typical values.

| Parameter | Symbol | Value | Unit |
|---|---|---|---|
| Salinity coefficient of freezing equation | $a$ | $-0.0572$ | $^{\circ}\mathrm{C\,PSU}^{-1}$ |
| Constant coefficient of freezing equation | $b$ | $0.0788$ | $^{\circ}\mathrm{C}$ |
| Pressure coefficient of freezing equation | $c$ | $7.77 \times 10^{-8}$ | $^{\circ}\mathrm{C\,Pa}^{-1}$ |
| Thermal expansion coefficient in EOS | $\alpha$ | $7.5 \times 10^{-5}$ | $^{\circ}\mathrm{C}^{-1}$ |
| Salt contraction coefficient in EOS | $\beta$ | $7.7 \times 10^{-4}$ | $\mathrm{PSU}^{-1}$ |
| Reference density in EOS | $\rho_*$ | $1033$ | $\mathrm{kg\,m}^{-3}$ |
| Latent heat of fusion | $L$ | $3.34 \times 10^{5}$ | $\mathrm{J\,kg}^{-1}$ |
| Heat capacity of sea water | $c_p$ | $3{,}974$ | $\mathrm{J\,kg}^{-1}\,{}^{\circ}\mathrm{C}^{-1}$ |
| Density of ice | $\rho_i$ | $910$ | $\mathrm{kg\,m}^{-3}$ |
| Density of sea water | $\rho_w$ | $1028$ | $\mathrm{kg\,m}^{-3}$ |
| Turbulent salinity exchange velocity | $\gamma_S$ | $2 \times 10^{-6}$ | $\mathrm{m\,s}^{-1}$ |
| Turbulent temperature exchange velocity | $\gamma_T$ | $5 \times 10^{-5}$ | $\mathrm{m\,s}^{-1}$ |
| Effective turbulent temperature exchange velocity | $\gamma_T^*$ | $2 \times 10^{-5}$ | $\mathrm{m\,s}^{-1}$ |
| Overturning strength | $C$ | $1 \times 10^{6}$ | $\mathrm{m}^3\,\mathrm{s}^{-1}$ |

The coefficients in the equation of state (EOS), the turbulent exchange velocities for heat and salt are taken from Olbers and
Hellmer (2010). We linearized the potential freezing temperature equation with a least-squares fit with salinity values over a
range of 20-40 PSU and pressure values of 0-$10^7$ Pa using Gibbs SeaWater Oceanographic Package of TEOS-10
(McDougall and Barker, 2011). All values are kept constant, except for $\gamma_T^*$ and $C$, which vary between experiments. The
values of these two parameters are the best fit values analysed in Sect. 3.1.

and compare the resulting sub-shelf melt rates to observational data. This is followed by a discussion of the applicability and
limitations of the model (Sect. 4) and conclusions (Sect. 5).

## 2 Model description

PICO is developed from the ocean box model of Olbers and Hellmer (2010), henceforth OH10. The OH10 model is designed
5 to capture the basic overturning circulation in ice shelf cavities which is driven by the "ice pump" mechanism: melting at the
ice shelf base near the grounding line reduces salinity and the ambient ocean water becomes buoyant, rising along the ice shelf
base towards the calving front. Since the ocean temperatures on the Antarctic continental shelf are generally close to the local
freezing point, density variations are primarily controlled by salinity changes. Melting at the ice-shelf base hence reduces the
density of ambient water masses, resulting in a haline-driven circulation. Buoyant water rising along the shelf base draws in
10 ocean water at depth, which flows across the continental shelf towards the deep grounding lines of the ice shelves. The warmer
these water masses are, the stronger is the melting-induced ice pump. The OH10 box model describes the relevant physical
processes and captures this vertical overturning circulation by defining consecutive boxes following the flow within the ice
shelf cavity.




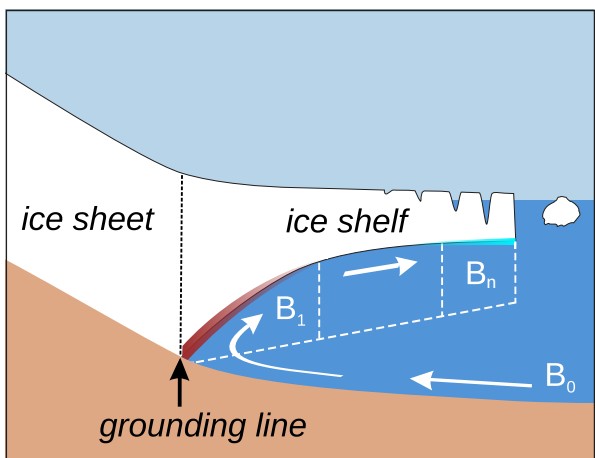

**Figure 1.** Schematic view of the PICO model. The model mimics the overturning circulation in ice shelf cavities: Ocean water from box $B_0$ enters the ice shelf cavity at the depth of the sea floor and is advected to the grounding line box $B_1$. Freshwater influx from melting at the ice shelf base makes the water buoyant, causing it to rise. The cavity is divided into $n$ boxes along the ice shelf base. Generally, the highest melt rates can be found near the grounding line, with lower melt rates or refreezing towards the calving front.

The strength of the overturning flux $q$ is determined from the density difference between the incoming water masses on the continental shelf and the buoyant water masses near the deep grounding lines of the ice shelf. As PICO is implemented in an ice sheet model with characteristic time scales much slower than typical response times of the ocean, we assume steady-state ocean conditions and hence reduce the complexity of the governing equations of the OH10 model. We assume stable vertical strati-

5  fication, which motivates neglecting the diffusive heat and salt transport between boxes[2]. Without diffusive transport between the boxes, some of the original ocean boxes from OH10 become passive and can be incorporated into the governing equations of the set of boxes used in PICO. We explicitly model a single open ocean box which provides the boundary conditions for the boxes adjacent to the ice shelf base following the overturning circulation, as shown in Fig. 1. In order to better resolve the complex melt patterns, PICO adapts the number of boxes based on the evolving geometry of the ice shelf. These simplifying

10  assumptions allow us to analytically solve the system of governing equations, which is presented in the following two sections. A detailed derivation of the analytic solutions is given in Appendix A. In Sect. 2.3, we describe how the ice-model grid relates to the ocean box geometry of PICO. The system of equations is solved locally on the ice-model grid, as described in Sect. 2.4. Table 1 summarizes the model parameters and typical values.

---

[2]OH10 discuss a circulation state for an unstable vertical water column, which would imply a high (parametrized) diffusive transport between boxes. They find that this state only occurs transiently (Olbers and Hellmer, 2010, Sect. 2).



## 2.1 Physics of the overturning circulation in ice shelf cavities

PICO solves for the transport of heat and salt between the ocean boxes as depicted in Fig. 1. Although box $B_0$, which is located at depth between the ice shelf front and the edge of the continental shelf, does not extend into the shelf cavity, its properties are transported unchanged from box $B_0$ to box $B_1$ near the grounding line. The heat and salt balances for all boxes in contact

with the ice shelf base (boxes $B_k$ for $k \in \{1, \ldots n\}$) can be written as

$$V_k \dot{T}_k = q T_{k-1} - q T_k + A_k m_k T_{bk} - A_k m_k T_k + A_k \gamma_T (T_{bk} - T_k) \tag{1}$$

$$V_k \dot{S}_k = q S_{k-1} - q S_k + A_k m_k S_{bk} - A_k m_k S_k + A_k \gamma_S (S_{bk} - S_k). \tag{2}$$

The local application of these equations for each ice model cell is described in Sect. 2.4. Since we assume steady circulation, the terms on the left-hand side are neglected. For the box $B_k$ with volume $V_k$, heat or salt content change due to advection

from the adjacent box $B_{k-1}$ with overturning flux $q$ (first term on the right-hand side) and due to advection to the neighboring box $B_{k+1}$ (or the open ocean for $k = n$) with overturning flux $q$ (second term). Vertical melt flux into the box $B_k$ across the ice-ocean interface with area $A_k$ (third term) and out of the box (fourth term) play a minor role and are neglected in the analytic solution of the equation system employed in PICO (a detailed discussion of these terms is given in Jenkins et al., 2001). The melt rate $m_k$ is negative if ambient water freezes to the shelf base. The last term represents heat and salt changes via turbulent,

vertical diffusion across the boundary layer beneath the ice-ocean interface. The parameters $\gamma_T$ and $\gamma_S$ are the turbulent heat and salt exchange velocities which we assume to be constant.

The overturning flux $q > 0$ is assumed to be driven by the density difference between the ocean reservoir box $B_0$ and the grounding line box $B_1$. This is parametrized as

$$q = C(\rho_0 - \rho_1) \tag{3}$$

where $C$ is a constant overturning coefficient that captures effects of friction, rotation and bottom formstress[3]. The circulation strength in PICO is hence determined by density changes through sub-shelf melting in the grounding zone box $B_1$. From there, water follows the ice shelf base towards the open ocean assuming the overturning flux $q$ to be the same for all subsequent boxes. Ocean water densities are computed assuming a linear approximation of the equation of state

$$\rho = \rho_*(-\alpha T + \beta S) \tag{4}$$

where $\alpha$, $\beta$ and $\rho_*$ are constants with values given in Table 1.

## 2.2 Melting physics

Melting physics are derived from the widely used 3-equation model (Hellmer and Olbers, 1989; Holland and Jenkins, 1999) which assumes the presence of a boundary layer below the ice-ocean interface. The temperature at this interface in box $B_k$ is assumed to be at the in-situ freezing point $T_{bk}$, which is linearly approximated by

$$T_{bk} = a S_{bk} + b - c p_k, \tag{5}$$

---

[3]For a more detailed discussion see Olbers and Hellmer (2010, Sect. 2).

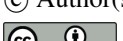



where $p_k$ is the overburden pressure, here calculated as static-fluid pressure given by the weight of the ice on top. At the ice-ocean interface, the heat flux from the ambient ocean across the boundary layer due to turbulent mixing, $Q_T = \rho_w c_p \gamma_T (T_{bk} - T_k)$, equals the heat flux due to melting or freezing $Q_{Tb} = -\rho_i L m_k$. We here neglect heat flux into the ice, the heat balance equation thus reads

$$\gamma_T (T_{bk} - T_k) = -\nu \lambda m_k \tag{6}$$

where $\nu = \rho_i / \rho_w \sim 0.89$, $\lambda = L/c_p \sim 84\,^\circ\mathrm{C}$. We obtain the salt flux boundary condition as the balance between turbulent salt transfer across the boundary layer, $Q_S = \rho_w \gamma_S (S_{bk} - S_k)$, and reduced salinity due to melt water input, $Q_{Sb} = -\rho_i S_{bk} m_k$,

$$\gamma_S (S_{bk} - S_k) = -\nu S_{bk} m_k. \tag{7}$$

To compute melt rates, we apply a simplified version of the 3-equations model (McPhee, 1992, 1999; Holland and Jenkins, 1999) which allows for a simple, analytic solution of the system of governing equations. It has been shown that this formulation yields realistic heat fluxes (McPhee, 1992, 1999). This simplification is used only for melt rates, the 3-equations formulation is applied from then on, with details laid out in Appendix A. Melt rates are given by

$$m_k = -\frac{\gamma_T^*}{\nu \lambda} (a S_k + b - c p_k - T_k) \tag{8}$$

with ambient ocean temperature $T_k$ and salinity $S_k$ in box $B_k$. Here, we use the effective turbulent heat exchange coefficient $\gamma_T^*$. The relation between $\gamma_T$ and $\gamma_T^*$ is discussed in the Appendix A.

## 2.3 PICO ocean box geometry

PICO is implemented as a module in the three-dimensional ice sheet model PISM as described in Sect. 2.4. Since the original system of box-model equations is formulated for only one horizontal and one vertical dimension, it needed to be extended for the use in the three-dimensional ice sheet model. To this aim, PICO distinguishes *basins*, which are chosen to encompass large ice shelf embayments and areas of similar ocean conditions. The standard basins used for Antarctica are shown in Fig. 2, see Sect. 3. The system of governing equations as described in the previous two sections are solved for each basin independently.

For any basin $D$, we determine the number of ocean boxes $n_D$ based on the size and geometry of the ice shelves such that larger ice shelves are resolved with more boxes. The number of boxes is defined separately for each basin by interpolating between 1 and $n_{\max}$ depending on the geometry of the ice shelves within that basin. The maximum number of boxes $n_{\max}$ is a model parameter; a value of 5 is used for the Antarctic setup, as discussed further in Sect 3.2. We determine the number of boxes $n_D$ for the basin $D$ with

$$n_D = 1 + \mathrm{rd}\left( \sqrt{d_{\mathrm{GL}}(D)/d_{\max}}\,(n_{\max} - 1) \right) \tag{9}$$

where rd rounds to the nearest integer. Here, $d_{\mathrm{GL}}(x,y)$ is the local distance to the grounding line from an ice-model grid cell with horizontal coordinates $(x,y)$, $d_{\mathrm{GL}}(D)$ the maximum distance within basin $D$ and $d_{\max}$ the maximum distance to the grounding line in the entire domain. PICO adapts the ocean boxes to the evolving ice shelves at every time step.



Knowing the maximum number of boxes $n_D$ for a basin $D$, we next define the ocean boxes underneath the ice shelves within that basin. The extent of boxes $B_1, \ldots, B_{n_D}$ is determined using the distance to the grounding line and the shelf front. The non-dimensional *relative distance to the grounding line* $r$ is defined as

$$r(x,y) = d_{\mathrm{GL}}(x,y) / (d_{\mathrm{GL}}(x,y) + d_{\mathrm{IF}}(x,y)) \tag{10}$$

with $d_{\mathrm{IF}}(x,y)$ the horizontal distance to the ice front. We assign all ice cells with horizontal coordinates $(x,y) \in D$ to box $B_k$ if the following condition is met

$$1 - \sqrt{(n_D - k + 1)/n_D} \leq r(x,y) \leq 1 - \sqrt{(n_D - k)/n_D}. \tag{11}$$

This leads to comparable areas for the different boxes within a basin, which is motivated in Appendix B. Thus, for example, the box $B_1$ adjacent to the grounding line interacts with all ice shelf grid cells with $0 \leq r \leq 1 - \sqrt{(n_D - 1)/n_D}$. Figure 3 shows an example of the ocean box areas for Antarctica.

## 2.4 Implementation in the Parallel Ice Sheet Model

PICO is implemented in the open-source Parallel Ice Sheet Model (PISM: Bueler and Brown, 2009; Winkelmann et al., 2011). In the 3d, thermo-mechanically coupled, finite-difference model, ice velocities are computed through a superposition of the shallow approximations for the slow, shear-dominated flow in ice sheets (Hutter, 1983, SIA) and the fast, membrane-like flow in ice streams and ice shelves (Morland, 1987, SSA). In PISM, the grounding lines (diagnosed via the flotation criterion) and ice fronts evolve freely. Grounding line movement has been evaluated in the model intercomparison project MISMIP3d (Pattyn et al., 2013; Feldmann et al., 2014).

Time-stepping in PICO is the same as in the ice-sheet model. The cavity model provides sub-shelf melt rates and temperatures at the ice-ocean boundary to PISM, with temperatures being at the in-situ freezing point. PISM supplies the evolving ice-shelf geometry to PICO, which in turn updates in each time step the ocean box geometry to the ice-shelf geometry as described in Sect. 2.3.

PICO computes the melt rates progressively over the ocean boxes, independently for each basin. Since the ice-sheet model has a much higher resolution, each ocean box interacts with a number of ice shelf grid cells. PICO applies the analytic solutions of the system of governing equations summarized in Sect. 2.1 and 2.2 locally to the ice model grid as detailed below. Model parameters that are varied between the experiments are the effective turbulent heat exchange velocity $\gamma_T^*$ from the melt parametrization described in Sect. 2.2 and the overturning coefficient $C$ described in Sect. 2.1. Despite the distinction into basins, the same parameter values are applied for the entire ice-sheet.

Input for PICO in the ocean reservoir box $B_0$ is data from observations or large-scale ocean models in front of the ice shelves. Temperature $T_0$ and salinity $S_0$ are averaged at the depth of the bathymetry in the continental shelf region. In box $B_1$ adjacent to the grounding line, PICO solves the system of governing equations in each ice grid cell $(x,y)$ to attain the overturning flux $q(x,y)$, temperature $T_1(x,y)$, salinity $S_1(x,y)$ and the melting $m_1(x,y)$ at its ice-ocean interface (given by the local solution of Eq. 3, Eq. A12, Eq. A8 and Eq. 8, respectively). The model proceeds progressively from box $B_k$ to box

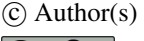



$B_{k+1}$ to solve for sub-shelf melt rate $m_{k+1}(x,y)$, ambient ocean temperature $T_{k+1}(x,y)$ and salinity $S_{k+1}(x,y)$ (given by the local solution of Eqn. 13, Eqn. A13 and Eqn. A8, respectively) based on the previous solutions $S_k$ and $T_k$ in box $B_k$ and conditions at the ice-ocean interface. PICO provides the boundary conditions $T_k$ and $S_k$ to box $B_{k+1}$ as the average over the ice-grid cells along the boundary between boxes $B_k$ and $B_{k+1}$ ensuring a smooth transition of sub-shelf melt rates and ocean

water properties, *i.e.*,

$$T_k = \langle T_k(x,y) \text{ with } (x,y) \text{ in } B_k \text{ and adjacent to } B_{k+1} \rangle \tag{12}$$

and analogously for $S_k$, where $\langle \ \rangle$ denotes the average.

The overturning is solved in Box $B_1$ and given by $q = \langle q(x,y) \text{ with } (x,y) \text{ in } B_1 \text{ and adjacent to } B_2 \rangle$. Melt rates in box $B_k$ are computed using the local overburden pressure $p_k(x,y)$ in each ice shelf grid cell that is given by the weight of the ice

column provided by PISM, *i.e.*,

$$m_k(x,y) = -\frac{\gamma_T^*}{\nu\lambda}(aS_k(x,y) + b - cp_k(x,y) - T_k(x,y)). \tag{13}$$

This reflects the pressure dependence of heat available for melting and leads to a depth-dependent melt rate pattern within each box. The implications for energy and mass conservation are discussed in Sect. 3.2 and Sect. 4.

## 3 Results for present-day Antarctica

We apply PICO to compute sub-shelf melt rates for all Antarctic ice shelves under present-day conditions. Based on Zwally et al. (2012), we determine 19 basins of the Antarctic Ice Sheet and extend these to the attached ice shelves and the surrounding Southern Ocean (Fig. 2). We combine drainage sectors feeding the same ice shelf, *e.g.*, all contributory inlets to Filchner-Ronne or Ross Ice Shelves. We also consolidate the basins 'IceSat21' and 'IceSat22' (Pine Island Glacier and Thwaites Glacier) as well as 'IceSat7' and 'IceSat8' in East Antarctica. Ocean conditions in box $B_0$ are given by observations of temperature

(converted to potential temperature) and salinity (converted to practical salinity) of the water masses occupying the sea floor on the continental shelf (Schmidtko et al., 2014), averaged over the time period 1975 to 2012. Figure 2 shows the basin-mean ocean temperature (shadings and numbers) and salinity (numbers) used as input values. Here, we use $n_{\max} = 5$ from which PICO determines the number of ocean boxes in each basin via Eq. 9. Figure 3 displays the resulting extent of the ocean boxes for Antarctica, ordered in elongated bands beneath the ice shelves. For the large ice-shelf cavities of Filchner-Ronne and

Ross the ice-ocean boundary is divided into five ocean boxes while smaller ice shelves have two to four boxes (see Table 2). Introducing more than five ocean boxes has a negligible effect on the melt rates, as discussed in Sect. 3.2.

To validate our model, we run diagnostic simulations with PISM+PICO based on bed topography and ice thickness from BEDMAP2 (Fretwell et al., 2013) mapped to a grid with $5\,\mathrm{km}$ horizontal resolution. Diagnostic simulations allow us to asses the sensitivity of the model to the parameters $C$ and $\gamma_T^*$ and to the number of boxes $n_{\max}$ as well as the ice model resolution.



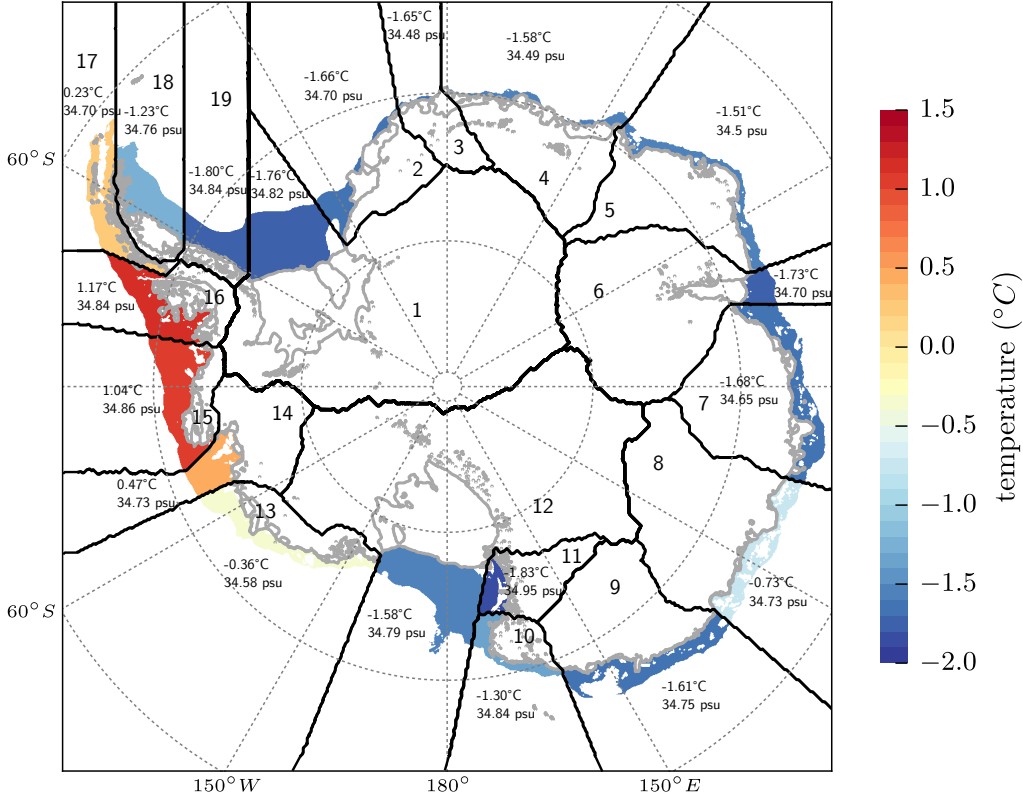

**Figure 2.** PICO input for Antarctic basins. The ice sheet, ice shelves and the surrounding Southern Ocean are split into 19 basins that are based on Zwally et al. (2012) and indicated by black contour lines and labels. For each basin, the governing equations are solved separately with the respective oceanic boundary conditions. Numbers show the temperature and salinity input in box $B_0$, obtained by averaging observed properties of the ocean water in front of the ice shelf cavities at depth of the continental shelf (Schmidtko et al., 2014), indicated by the color shading. Grey lines show Antarctic grounding lines and ice shelf fronts (Fretwell et al., 2013).

## 3.1 Sensitivity to model parameters $C$ and $\gamma_T^*$

We test the sensitivity of sub-shelf melt rates to the model parameters for overturning strength $C \in [0.1, 9]$ Sv and the effective turbulent heat exchange velocity $\gamma_T^* \in [5 \times 10^{-6}, 1 \times 10^{-4}]$ m s$^{-1}$. These ranges encompass the values identified in OH10, discussed further in Appendix A. The same parameters for $C$ and $\gamma_T^*$ are applied to all basins. We validate the results by the following sieve criteria, summarized in Fig. 4:

*Criterion (1):* Freezing must not occur in the first box $B_1$ of any basin, *i.e.*, the ocean box closest to the grounding line. Freezing in box $B_1$ would increase ambient salinity, and since the overturning circulation in ice-shelf cavities is mainly haline-driven, the circulation would shut down, violating the model assumption $q > 0$ (see Sect. 2). As shown in the upper left panel of Fig. 4, the condition is not met for a combination of relatively high turbulent heat exchange and relatively low overturning





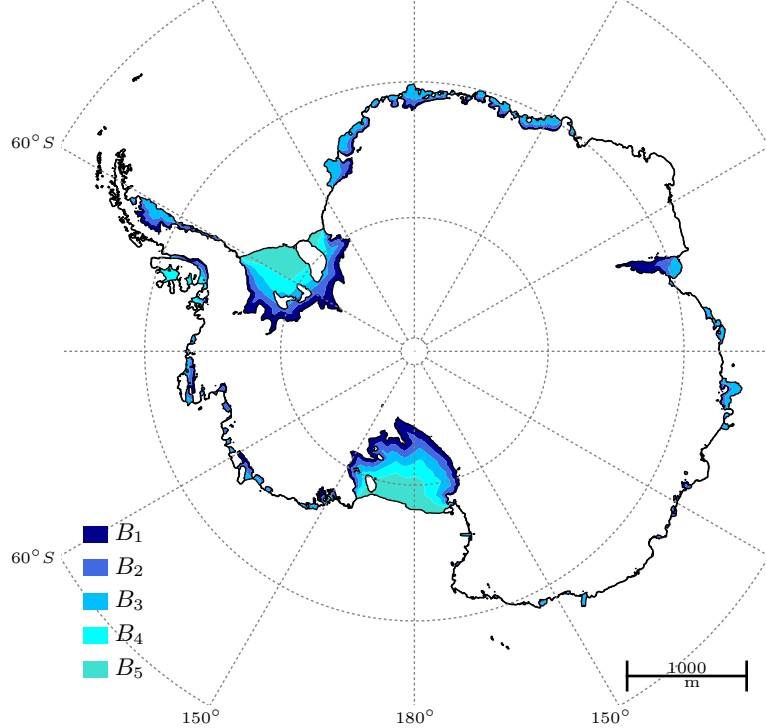

**Figure 3.** Extent of PICO ocean boxes for Antarctic ice shelves. Most ice shelves are split into two, three or four ocean boxes interacting with the ice cells on a much higher resolution. The largest ice shelves, Filchner-Ronne and Ross, have five ocean boxes. One ocean box typically corresponds to many ice shelf grid cells.

parameters. In such cases, freezing near the grounding line occurs because of the strong heat exchange between the ambient ocean and the ice-ocean boundary in box $B_1$ that cannot be balanced by the resupply of heat from the open ocean through overturning.

*Criterion (2):* Sub-shelf melt rates must decrease between the first and second box for each basin. This condition is based
5  on general observations of melt-rate patterns and on the assumption that ocean water masses move consecutively through the boxes and cool down along the way, as long as melting in these boxes outweighs freezing. As shown in the lower left panel in Fig. 4, this condition is violated for either high overturning and low turbulent heat exchange or, vice versa, low overturning and high turbulent heat exchange. An appropriate balance between the strength of these values is hence necessary for a realistic melt rate pattern.

10   If criterion (1) or (2) fails, basic assumptions of PICO are violated. Thus, we choose the model parameters $\gamma_T^*$ and $C$ such that both criteria are strictly met. The following quantitative criteria (3) and (4) compare modeled average melt rates with observations and thus depend on our choice of valid ranges. We choose basin 1 and 14 to further constrain our model parameters for Antarctica. Basin 1 contains the Filchner-Ronne Ice Shelf and basin 14 the ice shelves in the Amundsen Sea. These two basins represent two different types of ice shelves regarding both the mode of melting and the ice-shelf size.





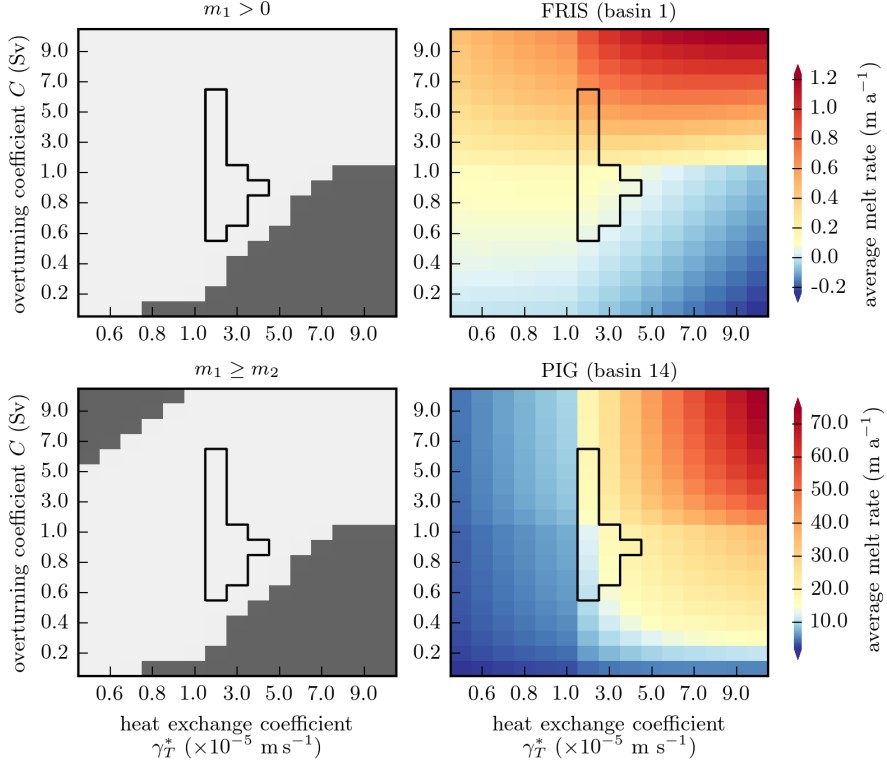

**Figure 4.** Sensitivity of PICO sub-shelf melt rates to the overturning coefficient $C$ and the effective turbulent heat exchange coefficient $\gamma_T^*$. Black contour indicates the valid range of parameters, all other parameter combinations are excluded by one of the following criteria: (Upper left) No freezing may occur in the first ocean box, (Lower left) mean basal melt rates must decrease between the first and second ocean box, (Upper right) mean basal melt rates for Filchner-Ronne Ice Shelf should be within the range of 0.05 to $1.0\,\mathrm{m\,a^{-1}}$, (Lower right) mean basal melt rates for the basin containing Pine Island Glacier should be within the range of 10 to $20\,\mathrm{m\,a^{-1}}$.

*Criterion (3):* Average melt rates in Filchner-Ronne Ice Shelf comply with the classification of a "cold" cavity and lie between 0.05 and $1.0\,\mathrm{m\,a^{-1}}$ (Fig. 4, upper right panel).

*Criterion (4):* In the Amundsen basin, for "warm" ocean conditions, average melt rates lie between 10 and $20\,\mathrm{m\,a^{-1}}$ (Fig. 4, lower right panel).

5    Generally, an increase in overturning strength $C$ will supply more heat and thus yield higher melt rates, especially for the large and "cold" ice shelves like Filchner-Ronne. In the valid parameter range, larger $C$ leads to higher melt rates also in the smaller and "warm" basins like Pine-Island but the effect is less pronounced. In contrast, the turbulent heat exchange alters melting particularly in basins with small ice shelves while it might decrease melt rates in large ice shelves with "cold" cavities. Hence, modeled melting in the Filchner-Ronne basin is dominated by overturning while in the Amundsen region melting is





dominated by turbulent exchange across the ice-ocean boundary layer. For three different parameter combinations, the resulting spatial patterns of melt rates in the Filchner-Ronne and Pine Island regions are displayed in Fig. S.1.

All of the above criteria restrict the parameter space to a bounded set with lower and upper limits as depicted by the contour line in Fig. 4. The valid range of model parameters with $C$ around $1\,\mathrm{Sv}$ and $\gamma_T^*$ around $2 \times 10^{-5}\,\mathrm{m\,a^{-1}}$ compares well with
5  those found in OH10 and Holland and Jenkins (1999).

## 3.2  Diagnostic melt rates for present-day Antarctica

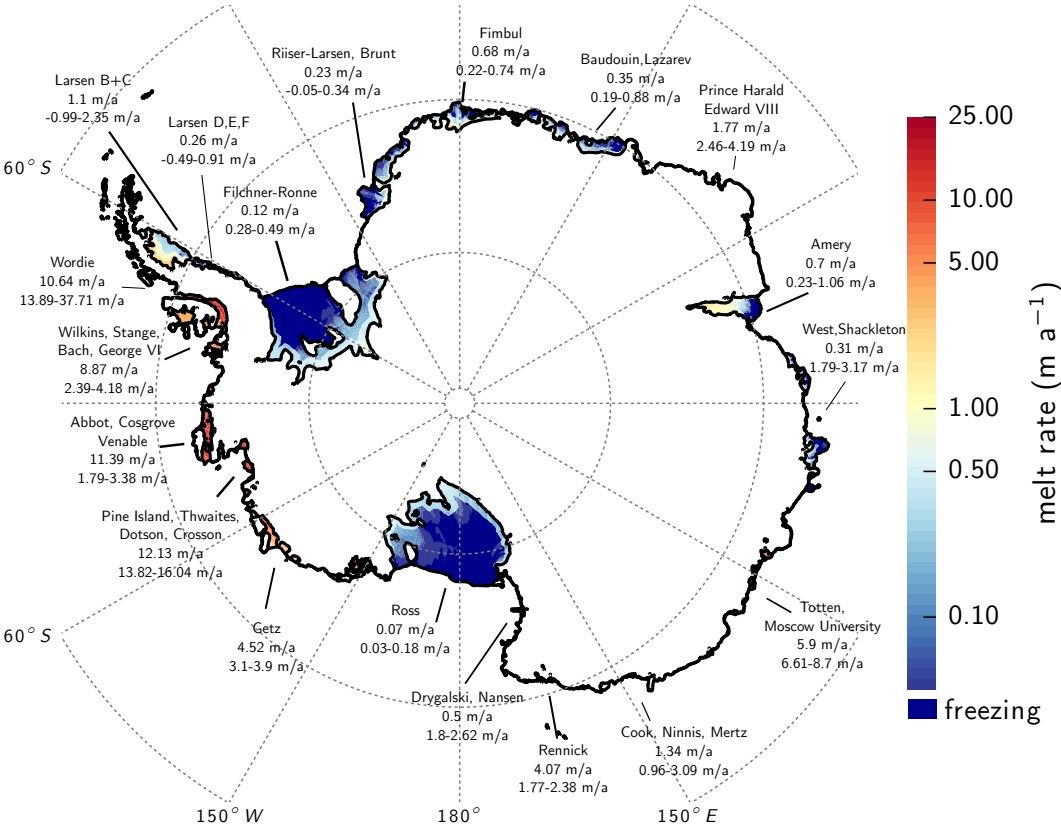

**Figure 5.** Sub-shelf melt rates for present-day Antarctica computed with PISM+PICO. For each basin, the mean melt rate (upper numbers) is compared to the observed range (lower numbers) from Rignot et al. (2013). In the model, the same parameters $\gamma_T^* = 2 \times 10^{-5}\,\mathrm{m\,s^{-1}}$ and $C = 1\,\mathrm{Sv}$ are applied to all ice shelves around Antarctica. The respective oceanic boundary condition are shown in Fig. 2. Ice geometry and bedrock topography are from the BEDMAP2 data set on $5\mathrm{km}$ resolution (Fretwell et al., 2013). Refreezing occurs in some parts of the larger shelves like Filchner-Ronne and Ross.

Using the best-fit values $C = 1\,\mathrm{Sv}$ and $\gamma_T^* = 2 \times 10^{-5}\,\mathrm{m\,s^{-1}}$ found in Sect. 3.1, we apply PICO to present-day Antarctica, solving for sub-shelf melt rates and water properties in the ocean boxes. This model simulation is referred to as "reference simulation" hereafter.





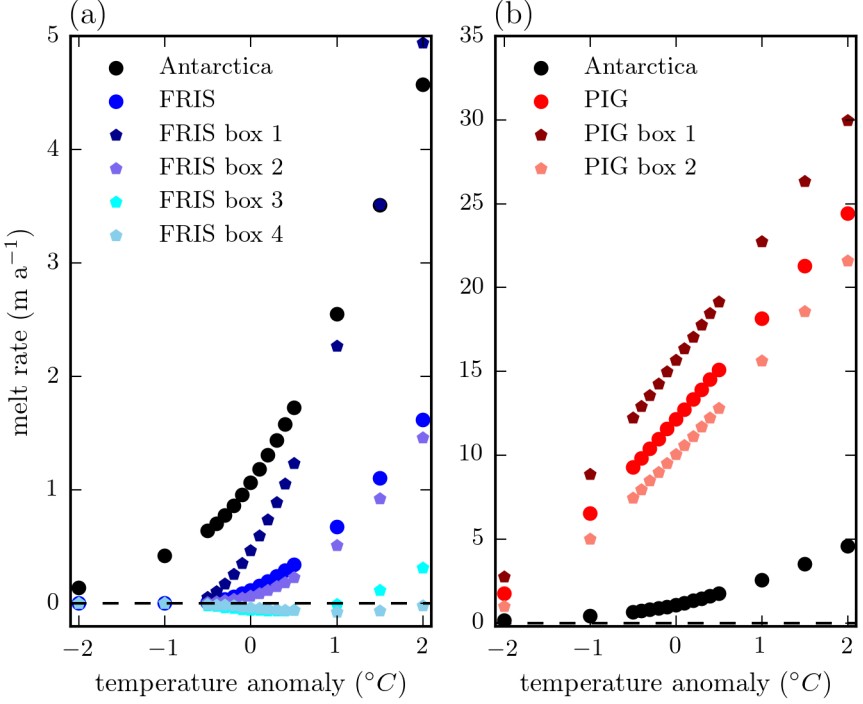

**Figure 6.** Sensitivity of PICO sub-shelf melt rates to ocean temperature changes for entire Antarctica (black), Filchner-Ronne Ice Shelf (blue) and the basin containing Pine Island Glacier (red). Ocean input temperatures are varied by $0.1°C$ up to $2°C$. Melting depends quadratically on temperature for "cold" cavities like the one adjacent to Filchner-Ronne, and linearly for "warm" cavities like the ones in the Amundsen Region.

The average melt rates computed with PICO range from $0.07\,\mathrm{m\,a^{-1}}$ under the Ross Ice Shelf to $12.13\,\mathrm{m\,a^{-1}}$ for the Amundsen Region (Fig. 5). Generally, melt rates are highest in the vicinity of the grounding line and decrease towards the calving front. In some regions of the large ice shelves, refreezing occurs, *e.g.*, towards the center of Filchner-Ronne or Amery ice shelves. The melt pattern also depends on the local pressure melting temperature, which is a function of the local ice thickness. Thus, in some boxes freezing occurs in regions of relatively thin ice while melting occurs in regions where the ice shelf is thicker. For the vast majority of ice shelves, the modeled average melt rates compare well with the observed ranges derived from Table 1 in Rignot et al. (2013). An exception are the two basins containing Abbot and Cosgrove ice shelves (basin 15) as well as Wilkins and Stange ice shelves (basin 16) with average modeled melt rates of $11.39\,\mathrm{m\,s^{-1}}$ and $8.87\,\mathrm{m\,s^{-1}}$ respectively, which is much higher than the observed range of $1.8-3.4\,\mathrm{m\,s^{-1}}$ and $2.39-4.18\,\mathrm{m\,s^{-1}}$. This is most likely due to the ocean temperature input for these basins ($1.04\,°C$ and $1.16\,°C$, see Fig. 2) which is higher than for the basin containing Pine Island located nearby ($0.47°C$, basin 14), explaining why the melt rates are of the same order of magnitude in these basins. Modifica-



**Table 2.** Results from the reference simulation as displayed in Fig. 5.

| | basin | $b_n$ | $T_0$ | $S_0$ | $T_n$ | $S_n$ | $\Delta T$ | $\Delta S$ | $q$ | $m$ | $m_{observed}$ |
|---|---|---|---|---|---|---|---|---|---|---|---|
| Filchner-Ronne | 1 | 5 | -1.76 | 34.65 | -2.13 | 34.48 | -0.37 | -0.17 | 0.19 | 0.12 | 0.28-0.49 |
| Riiser-Larsen, Brunt | 2 | 3 | -1.66 | 34.53 | -2.02 | 34.36 | -0.36 | -0.17 | 0.09 | 0.23 | -0.05-0.34 |
| Fimbul | 3 | 3 | -1.58 | 34.32 | -2.03 | 34.12 | -0.45 | -0.21 | 0.10 | 0.68 | 0.22-0.74 |
| Lazarev, Baudouin | 4 | 3 | -1.55 | 34.33 | -2.01 | 34.11 | -0.47 | -0.22 | 0.12 | 0.35 | 0.19-0.88 |
| Prince Harald, Edward VIII | 5 | 3 | -1.51 | 34.33 | -1.90 | 34.15 | -0.38 | -0.18 | 0.08 | 1.77 | 2.46-4.19 |
| Amery | 6 | 3 | -1.72 | 34.53 | -2.08 | 34.37 | -0.35 | -0.16 | 0.12 | 0.70 | 0.23-1.06 |
| West, Shackleton | 7 | 3 | -1.69 | 34.48 | -2.00 | 34.34 | -0.31 | -0.14 | 0.09 | 0.31 | 1.79-3.17 |
| Totten, Moscow University | 8 | 2 | -0.68 | 34.57 | -1.56 | 34.16 | -0.88 | -0.41 | 0.20 | 5.90 | 6.61-8.70 |
| Cook, Ninnis, Mertz | 9 | 3 | -1.62 | 34.58 | -2.04 | 34.38 | -0.43 | -0.20 | 0.09 | 1.34 | 0.96-3.09 |
| Rennick | 10 | 2 | -1.31 | 34.67 | -1.69 | 34.49 | -0.38 | -0.18 | 0.08 | 4.07 | 1.77-2.38 |
| Drygalski, Nansen | 11 | 3 | -1.84 | 34.78 | -2.00 | 34.71 | -0.16 | -0.08 | 0.03 | 0.50 | 1.80-2.62 |
| Ross | 12 | 5 | -1.58 | 34.63 | -2.05 | 34.40 | -0.47 | -0.22 | 0.17 | 0.07 | 0.03-0.18 |
| Getz | 13 | 3 | -0.37 | 34.41 | -1.81 | 33.75 | -1.44 | -0.66 | 0.31 | 4.52 | 3.10-3.90 |
| PIG, Thwaites, Dotson, Crosson | 14 | 2 | 0.46 | 34.55 | -0.90 | 33.93 | -1.35 | -0.62 | 0.23 | 12.13 | 13.82-16.04 |
| Abbot, Cosgrove, Venable | 15 | 3 | 1.04 | 34.69 | -1.15 | 33.68 | -2.19 | -1.01 | 0.35 | 11.39 | 1.79-3.38 |
| Wilkins, Stange, Bach, George VI | 16 | 4 | 1.17 | 34.67 | -1.46 | 33.45 | -2.64 | -1.21 | 0.35 | 8.87 | 2.39-4.18 |
| Wordie | 17 | 2 | 0.23 | 34.53 | -0.03 | 34.41 | -0.26 | -0.12 | 0.06 | 10.64 | 13.89-37.71 |
| Larsen B,C | 18 | 3 | -1.23 | 34.58 | -1.99 | 34.23 | -0.76 | -0.35 | 0.17 | 1.10 | -0.99-2.35 |
| Larsen D,E,F | 19 | 2 | -1.79 | 34.66 | -1.93 | 34.60 | -0.14 | -0.06 | 0.04 | 0.26 | -0.49-0.91 |

Basins are labeled according to prominent ice shelves; $b_n$ is the number of boxes, $T_0$ ($S_0$) is the temperature (salinity) in ocean box $B_0$, $T_n$ ($S_n$) the temperature (salinity) averaged over the ocean box at the ice shelf front, $\Delta T = T_n - T_0$ and $\Delta S = S_n - S_0$. $m$ is the average sub-shelf melt rate, $m_{observed}$ the uncertainty range of observed melt rates calculated from Rignot et al. (2013). $q$ is the overturning flux. Unit of temperatures is $^\circ$C, salinity is given in PSU, melt rates in m a$^{-1}$ and overturning flux in Sv.

tion of water masses flowing into the shelf cavities, not captured by PICO, might explain the low observed melt rates in basins 15 and 16 despite the relatively high ocean temperatures.

For all basins, ocean temperatures and salinities consistently decrease in overturning direction, *i.e.,* from the ocean reservoir box $B_0$ to the last box adjacent to the ice front $B_n$, as shown in Table 2. Most basins contain small areas in which ocean water freezes to the ice shelf base, with a maximum rate of $-0.63\,\mathrm{m\,s}^{-1}$ for the Amery Ice Shelf, see Table S.1. No freezing occurs at the Western Antarctic Peninsula nor in the Amundsen and Bellingshausen Seas. A detailed map of sub-shelf melt rates in this region as well as for Filchner-Ronne Ice Shelf can be found in Fig. S.1. For the Filchner-Ronne Ice Shelf melt rates vary between $-0.49\,\mathrm{m\,a}^{-1}$ and $1.76\ \mathrm{m\,a}^{-1}$ and for the basin containing Pine Island Glacier, melt rates range from $8.87\,\mathrm{m\,a}^{-1}$ to $18.85\,\mathrm{m\,a}^{-1}$.

Aggregated over all Antarctic ice shelves, the total melt flux is $1,299\,\mathrm{Gt\,a}^{-1}$, close to the observed estimate of $1,500 \pm 237$ $\mathrm{Gt\,a}^{-1}$ (Rignot et al., 2013). Overturning fluxes in our reference simulation range from $0.03\,\mathrm{Sv}$ for the basin containing the




small ice shelves Drygalski and Nansen to 0.35 Sv along the Western Antarctic Peninsula. These overturning fluxes compare well with the estimates in OH10.

PICO solves the system of governing equations locally in each ice-model grid cell and calculates input for each ocean box as an average along the boxes boundary as described in Sect. 2.4. Due to this model assumptions, mass and energy are a-priori not
perfectly conserved. In Table S.1, we compare (within each basin) heat fluxes into the ice shelf cavities with the heat flux out of the cavities into the ocean and the latent heat flux for melting. For the whole of Antarctica, the deviation in heat flux is $-282.15$ $\mathrm{GJ\,s^{-1}}$ which is equivalent to $2.0\%$ of the latent heat flux for melting. The per-basin deviations are generally low ($< 15\%$), except around Amery and Filchner-Ronne ice shelves. This can be explained by an underestimation of the overturning in these particular basins, which is due to the computation of the overturning flux $q$ along the boundary between boxes $B_1$ and $B_2$ (at a
depth of $423\,\mathrm{m}$ and $700\,\mathrm{m}$, respectively) instead of using the average shelf depth in $B_1$ (which is $671\,\mathrm{m}$ and $839\,\mathrm{m}$). Summed over all Antarctic ice shelves, the error in overturning introduced by this choice of implementation is however small. In PICO we assume $q$ to be constant, neglecting changes due to melt water input along the shelf base. This melt water input amounts to $3.17\%$ or less of the overturning flux within each basin, and $1.4\%$ for the entire continent, discussed in Sect. 2.1.

Melt rates are strongly affected by changes in the ambient ocean temperatures, see Fig. 6. The dependence is approximately
linear for high and quadratic for lower ambient ocean temperatures. This relationship is similar to the one observed in OH10 and as expected from the governing equations. In Pine Island Glacier, melt rates increase by approximately $6\,\mathrm{m\,a^{-1}}$ per degree of warming. Changes in the ice-sheet model resolution have little effect on the resulting melt rates (Fig. S.2). For increasing the maximum number of boxes $n_{\mathrm{max}}$, average melt rates converge to almost constant values for $n_{\mathrm{max}} \geq 5$ within all basins, compare Fig. S.3.

## 4  Discussion

PICO models the dominant vertical overturning circulation in ice shelf cavities and translates ocean conditions in front of the ice shelves, either from observations or large-scale ocean models, into physically-based sub-shelf melt rates. For present-day ocean fields and ice-shelf cavity geometries, PICO as an ocean module in PISM reproduces average melt rates of the same order of magnitude as observations for all Antarctic basins. With a single combination of overturning parameter $C$ and
effective turbulent heat exchange parameter $\gamma_T^*$ applied to all basins, a wide range of melt rates for the different ice shelves is obtained, reproducing the large-scale patterns observed in Antarctica. The results are consistent across different ice-sheet and cavity model resolutions. Additionally, PICO reproduces the common pattern of maximum melt close to the grounding line and decreasing melt rates towards the ice shelf front, eventually with re-freezing in the shallow parts of the large ice shelves. The governing model equations are solved for individual grid cells of the ice sheet model (and not for each ocean box with
representative depth value), which yields a comparably high resolution of the obtained melt rate field.

In the underlying equations, transversal transport within the ice shelf cavities, *e.g.*, due to Coriolis force is not represented. Seasonal melt rate variation due to intrusion of warm water from the calving front during Austral summer is also not included in the model. Boundary conditions as input for the next-following ocean box are evaluated as mean along the inter-box boundary,





which permits a smooth distribution of modeled melt rates across the ice shelves. For the entire continent, the relative error in the overturning flux introduced by averaging along the boundary as compared to the mean over the entire ocean box is $3.5\%$. For the estimated heat fluxes, the relative error is lower than $2.0\%$ of the latent heat flux due to sub-shelf melting. Regarding mass conservation the relative error introduced by assuming the overturning to be constant along the boxes is below $1.4\%$ of

the total overturning strength. We hence consider our choice of model simplifications as justified regarding the associated small errors introduced in the heat and mass balances for our reference simulation.

In PICO, melt rates show a quadratic dependency on ocean temperature input for lower temperatures, *e.g.*, in the Filchner-Ronne basins, and a rather linear dependency for higher temperatures, *e.g.*, in the Amundsen basin. This is consistent with the results from OH10 and the implemented melting physics assuming a constant coefficient for turbulent heat exchange.

In contrast, Holland et al. (2008b) employ a dependency of the turbulent heat exchange coefficient on the velocity of the overturning circulation, suggesting melt rates to respond quadratically to warming of the ambient ocean water. Here we follow the approach taken in OH10.

PICO is computationally very fast, as it uses analytic solutions of the equations of motion with a small number of boxes along the ice shelf. As boundary conditions for PICO are aggregated based on predefined regional basins, the model can act as

an efficient coupler of large-scale ice-sheet and ocean models. For this purpose, heat flux into the ice should be added to the boundary layer melt formulation.

## 5    Conclusions

The Antarctic Ice Sheet plays a vital role in modulating global sea level. The ice grounded below sea level in its marine basins is susceptible to ocean forcing and responds nonlinearly to changes in ocean boundary conditions (Mercer, 1978; Schoof, 2007).

We therefore need carefully estimated conditions at the ice-ocean boundary to better constrain the dynamics of the Antarctic ice and its contribution to sea-level rise for the past and the future.

The PICO model presented here provides a physics-based yet efficient approach for estimating the ocean circulation below ice shelves and the heat provided for ice shelf melt. The model extends the one-horizontal-dimensional ocean box model by OH10 to realistic ice shelf geometries following the shape of the grounding line and calving front. PICO is a comparably

simple and fast alternative to full ocean models, but goes beyond local melt parameterizations, which do not fully reflect the circulation below ice shelves. We validated the model using present-day ocean conditions and ice geometries. PICO accurately reproduces the general pattern of ice shelf melt, with high melting at the grounding line and low melting or refreezing towards the calving front. Its sensitivity to changes in input ocean temperatures and model parameters is comparable to earlier estimates (Holland and Jenkins, 1999; Olbers and Hellmer, 2010). The model accurately captures the large variety of observed Antarctic

melt rates using only two calibrated parameters that are valid for the whole ice sheet.

The ocean models that are part of the large Earth system and global circulation models do not yet resolve the circulation below ice shelves. PICO is able to fill this gap and can be used as an intermediary between global circulation models and ice sheet models. We expect that PICO will be useful for providing ocean forcing to ice sheet models with the standardized input





from climate model intercomparison projects like CMIP5 and CMIP6 (Taylor et al., 2012; Meehl et al., 2014; Eyring et al., 2016). Since PICO can deal with evolving ice shelf geometries in a computationally efficient way, it is in particular suitable for modeling the ice sheet evolution on paleo-climate timescales as well as for future projections.

PICO is implemented as a module in the open-source Parallel Ice Sheet Model (PISM). The source code is fully accessible

and documented as we want to encourage improvements and implementation in other ice sheet models. This includes the adaption to other ice sheets than present-day Antarctica.

## 6  Code availability

The PICO code is part of the PISM-PIK development branch and openly available[4]. A merge into the general PISM stable version 08 is underway.

## Appendix A:  Derivation of the analytic solutions

Here, we derive the analytic solutions of the equations system describing the overturning circulation (see Sect. 2.1) and the melting at the ice-ocean interface (see Sect. 2.2).

For box $B_k$ with $k > 1$ we solve progressively for melt rate $m_k$, temperature $T_k$ and salinity $S_k$ in box $B_k$, dependent on the local pressure $p_k$, the area of box adjacent to the ice shelf base $A_k$ and the temperature $T_{k-1}$ and salinity $S_{k-1}$ of the upstream

box $B_{k-1}$. For box $B_1$, we additionally solve for the overturning $q$ as explained below. These derivations advance the ideas presented in the appendix of OH10. Assuming steady state conditions, the balance equations Eqs. 1 and 2 for box $B_k$ from Sect. 2.1 are

$$0 = q(T_{k-1} - T_k) + A_k \gamma_T (T_{bk} - T_k) + A_k m_k (T_{bk} - T_k)$$
$$0 = q(S_{k-1} - S_k) + A_k \gamma_S (S_{bk} - S_k) + A_k m_k (S_{bk} - S_k) \tag{A1}$$

The heat fluxes balance at the boundary layer interface, *i.e.*, the heat flux across the boundary layer due to turbulent mixing $Q_T = \rho_w c_p \gamma_T (T_{bk} - T_k)$ equals the heat flux due to melting or freezing $Q_{Tb} = -\rho_i L m_k$, omitting the heat flux into the ice. This yields

$$\gamma_T (T_{bk} - T_k) = -\nu \lambda m_k, \tag{A2}$$

where $\nu = \rho_i / \rho_w \sim 0.89$, $\lambda = L/c_p \sim 84°C$. Regarding the salt flux balance in the boundary layer, with $Q_S = \rho_w \gamma_S (S_{bk} - S_k)$

at the lower interface of the boundary layer and "virtual" salt flux due to meltwater input $Q_{Sb} = -\rho_i S_{bk} m_k$, we obtain

$$\gamma_S (S_{bk} - S_k) = -\nu S_{bk} m_k. \tag{A3}$$

---

[4]https://github.com/pism/pism





Inserting Eqs. A2 and A3 into Eqs. A1 yields

$$0 = q(T_{k-1} - T_k) - A_k m_k \nu \lambda + A_k m_k (T_{bk} - T_k)$$

$$0 = q(S_{k-1} - S_k) - A_k m_k \nu S_{bk} + A_k m_k (S_{bk} - S_k).$$

Comparing $(T_{bk} - T_K) << \nu \lambda \approx 75\,^\circ\mathrm{C}$, allows us to neglect the last term in the temperature equation. Considering the last

two terms of the salinity equation, we find that $S_k > (1-\nu)S_{bk} \approx 0.1\,S_{bk}$, allowing us to neglect the terms containing $S_{bk}$, which simplifies the equations to

$$0 = q(T_{k-1} - T_k) - A_k \nu \lambda m_k$$

$$0 = q(S_{k-1} - S_k) - A_k m_k S_k. \tag{A4}$$

We use a simplified version of the melt law described by McPhee (1992) and detailed in Sect. 2.2, which makes use of Eqn. 6

and Eqn. 5 in which the salinity in the boundary layer $S_{bk}$ is replaced by salinity of the ambient ocean water.

$$m_1 = -\frac{\gamma_T^*}{\nu \lambda}(aS_k + b - cp_k - T_k). \tag{A5}$$

Holland and Jenkins (1999) suggest that this simplification requires $\gamma_T^*$ to be a factor of 1.35 to 1.6 smaller than $\gamma_T$ in the 3-equation formulation for the constant values of $\gamma_T$ ranging from $3 \times 10^{-5}$ m s$^{-1}$ to $5 \times 10^{-5}$ m s$^{-1}$ used in OH10. This implies that $\gamma_T^*$ ranges from $2.2 \times 10^{-5}$ m s$^{-1}$ to $3.2 \times 10^{-5}$ m s$^{-1}$, which is consistent with the parameter range we derive in

Sect. 3.1. Inserting the simplified melt law in Eqs. A4 yields

$$0 = q(T_{k-1} - T_k) + A_k \frac{\gamma_T^*}{\nu \lambda}(aS_k + b - cp_k - T_k)\,\nu \lambda$$

$$0 = q(S_{k-1} - S_k) + A_k \frac{\gamma_T^*}{\nu \lambda}(aS_k + b - cp_k - T_k)\,S_1$$

Replacing $x = T_{k-1} - T_k$, $y = S_{k-1} - S_k$, $T^* = aS_{k-1} + b - cp_k - T_{k-1}$, $g_1 = A_k \gamma_T^*$ and $g_2 = \frac{g_1}{\nu \lambda}$, we obtain

$$0 = qx + g_1(T^* + x - ay) \tag{A6}$$

$$0 = qy + g_2(S_{k-1} - y)(T^* + x - ay) \tag{A7}$$

We simplify the previous equations as follows. Rewriting Eq. A6

$$(T^* + x - ay) = \frac{-qx}{g_1}$$

and inserting it into Eq. A7, we obtain

$$0 = qy + g_2(S_{k-1} - y)\left(-\frac{qx}{g_1}\right) = qy - qx\frac{S_{k-1} - y}{\nu \lambda}$$

$$\Longleftrightarrow 0 = \nu \lambda y - S_{k-1}x + xy$$

$$\Longleftrightarrow 0 = (\nu \lambda + x)y - S_{k-1}x$$

$$\Rightarrow y = \frac{S_{k-1}x}{\nu \lambda + x}.$$




Note that we can divide the first line by $q$ since, by the model assumptions, $q > 0$. Because $x = T_{k-1} - T_k << \nu\lambda \approx 75\,^\circ\mathrm{C}$, we may approximate

$$y \approx \frac{S_{k-1}x}{\nu\lambda}. \tag{A8}$$

Using this approximation, we may proceed to solve the system of equations. Since we also need to solve for the overturning $q$

in box $B_1$, which is adjacent to the grounding line, a slightly different approach is needed than for the other boxes, as discussed in the next section.

**Solution for box $B_1$**

The overturning flux $q$ is parameterized as

$$q = C\rho_* \left( \beta(S_0 - S_1) - \alpha(T_0 - T_1) \right), \tag{A9}$$

in the model, see Sect. 2.1. Substituting this equation into Eqs. A6 and A7, we obtain

$$0 = \alpha x^2 - \beta xy - \frac{g_1}{C\rho_*}(T^* + x - ay) \tag{A10}$$

$$0 = -\beta y^2 + \alpha xy - \frac{g_2}{C\rho_*}(S_0 - y)(T^* + x - ay). \tag{A11}$$

Inserting the approximation for $y$ from Eqn. A8 into the Eqn. A10, we obtain a quadratic equation for $x$,

$$(\beta s - \alpha)x^2 + \frac{g_1}{C\rho_*}(T^* + x(1 - as)) = 0$$

with $s = S_0/\nu\lambda$. Since $as = -0.057 \times S_0/74.76 = -0.000762 \times S_0 << 1$, we can omit the last part of the last term,

$$(\beta s - \alpha)x^2 + \frac{g_1}{C\rho_*}(T^* + x) = 0.$$

Rearranging (assuming that $\beta s - \alpha > 0$, which we demonstrate below), we obtain

$$x^2 + \frac{g_1}{C\rho_*(\beta s - \alpha)}x + \frac{g_1 T^*}{C\rho_*(\beta s - \alpha)} = 0,$$

and hence we obtain the solution

$$x = -\frac{g_1}{2C\rho_*(\beta s - \alpha)} \pm \sqrt{\left(\frac{g_1}{2C\rho_*(\beta s - \alpha)}\right)^2 - \frac{g_1 T^*}{C\rho_*(\beta s - \alpha)}}. \tag{A12}$$

The temperature in the box $B_1$ near the grounding line is supposed to be smaller than in the ocean box $B_0$, since, in general, melting will occur in box $B_1$ and hence $T_1 < T_0$, or equivalently $x = T_0 - T_1 > 0$. Furthermore, we know that $g_1/(C\rho_*) = A_1\gamma_T^*/(C\rho_*)$ is positive, as all factors are positive. Since $\alpha = 7.5 \times 10^{-5}$, $\beta = 7.7 \times 10^{-4}$ and $s = S_0/(\nu\lambda) = S_0/74.76 \geq 0.4$, it follows that $\beta s > \alpha$. This means that the first summand of Eqs. A12 is negative and the second (negative) solution can be

excluded. From here, we use $T_1 = T_0 + x$ and $y = xS_0/(\nu\lambda)$ to solve for $T_1$, $S_1$, $m_1$ and $q$.



**Solution for box $B_k$, $k > 1$**

Now, we give the solution for the other boxes $B_k$ with $k > 1$. By inserting the approximation for $y$ in Eqs A8 into Eq. A6, we can solve for $x$ as

$$0 = qx + g_1\left(T^* + x - a\frac{S_{k-1}x}{\nu\lambda}\right)$$

$$\iff 0 = qx + g_1 T^* + g_1 x - g_2\, a\, S_{k-1}\, x$$

$$\iff -g_1 T^* = x(q + g_1 - g_2\, S_{k-1}\, a)$$

$$\iff x = \frac{-g_1 T^*}{q + g_1 - g_2\, a\, S_{k-1}}. \tag{A13}$$

The denominator is positive, as all terms are positive, and the sign of the numerator depends on $T^*$. The equation can now be solved for $T_k$, and then Eqn. A8 for $S_k$ and Eqn. 13 for $m_k$.

## Appendix B: Motivation for geometric rule

Here, we want to motivate the rule that determines the extent of the boxes under each ice shelf. The rule aims at equal areas for all boxes. Assuming a half-circle with radius $r = 1$, we want to split it into a fixed number $n$ of equal-area rings. Generalized to the individual shapes of ice-shelf basins, we will define the "radius" of an ice shelf as $r = 1 - d_{\mathrm{GL}}/(d_{\mathrm{GL}} + d_{\mathrm{IF}})$. We define $r_1 = 1$ the outer (grounding-line ward) radius of the half-circle ring covering an area $A_1$ and corresponding to box $B_1$ adjacent to the grounding line, $r_2$ as the outer radius of second outer-most half-ring, etc. The box $B_k$ is then given by all shelf cells with horizontal coordinates $(x, y)$ such that $r_{k+1} \leq r(x,y) \leq r_k$ where $r_{n+1} = 0$ is the center point of the circle. We can use these to determine the areas $A_n = 0.5\pi r_n^2$, $A_{n-1} = 0.5\pi(r_{n-1}^2 - r_n^2)$, ..., $A_{n-k} = 0.5\pi(r_{n-k}^2 - r_{n-k+1}^2)$. If we require that $A_1 = A_2 = \cdots = A_n$, then, solving progressively, $r_{n-k} = \sqrt{k+1}\, r_n$. By our assumption is $r_1 = 1$, hence $1 = r_{n-(n-1)} = \sqrt{n}\, r_n$. This implies that $r_n = 1/\sqrt{n}$ and thus $r_{n-k} = \sqrt{\frac{k+1}{n}}$. Hence, the box $B_k$ for $k = 1, \ldots, n$ is defined as $1 - \sqrt{(n-k+1)/n} \leq d_{\mathrm{GL}}/(d_{\mathrm{GL}} + d_{\mathrm{IF}}) \leq 1 - \sqrt{(n-k)/n}$.

*Acknowledgements.* PISM development is supported by the NASA Modeling, Analysis, and Prediction program grant number NNX13AM16G and the NASA Cryospheric Sciences program grant number NNX16AQ40G and by NSF Polar Programs grants PLR-1603799 and PLR-1644277. T. A. was supported by DFG priority program SPP 1158, project numbers LE1448/6-1 and LE1448/7-1. M. M. was supported by the AXA Research Fund. The project is further supported by the German Climate Modeling Initiative (PalMod) and the Leibniz project DominoES. X. A. D. was supported by the US Department of Energy, Office of Science, Office of Biological and Environmental Research under award no. DE-SC0013038. The authors gratefully acknowledge the European Regional Development Fund (ERDF), the German Federal Ministry of Education and Research and the Land Brandenburg for supporting this project by providing resources on the high performance computer system at the Potsdam Institute for Climate Impact Research.



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
