# Peer review of "Antarctic sub-shelf melt rates via PICO"

_The Cryosphere, 2017_

## Referee Comment (RC1) · F. Pattyn (Referee) · 24 Jul 2017

**General comments**

This is a great paper that fills in the gap that is currently existing in linking large scale ocean to ice-sheet models. At this time, it is probably the best alternative to fully coupled ice-shelf - ocean cavity circulation in order to determine basal melt rates underneath ice shelves. The method is based on the Olbers and Hellmer box model (OH10), but extended to two plan-view dimensions. While it encompasses a series of approximation to this simple model, it is superior to current parametrizations used in large-scale ice-sheet modelling relating melt rates to ice draft. The paper is well written and gives sufficient details on how the model is derived from OH10 and implemented numerically.

[Figure]

The basic premise of the PICO ocean-coupler (if I may call it so) is that circulation in ice shelf cavities is based on vertical overturning. Parameters are therefore chosen in such a way that overturning is applied to all sub-shelf cavities around Antarctica, leading to sub-shelf melting close to the grounding line and accretion (if conditions apply) away from it. Results of the model applied to present-day Antarctic ice shelves gives sub-shelf melt rates close in agreement with observed values. A brief sensitivity analysis demonstrates the effect of ocean temperatures on sub-shelf melt rates.

While the model is definitely interesting for use as an ocean coupler (in absence of fully-coupled solutions), some care should be taken in its future use: it is based on stable vertical stratification, it only considers overturning circulation under ice shelves, it neglects Coriolis effects, and it relates ocean temperature (not circulation or intrusion of CDW underneath ice shelves) to sub-shelf melt. However, major advantages are that it considers the physics of th overturning circulation and that ice shelf size (given by the number of sub-shelf boxes) and distance to grounding line and ice shelf front matters.

I have only two major comments on the paper:

1. Why using basins and not individual ice shelves to link to mean values of $T_0$ and $S_0$? It seems to me that ocean circulation (and temperature/salinity) is related to individual ice shelves and not to drainage basins, which are governed by inland ice flow. Furthermore, during prognostic simulations, these drainage boundaries may change over time, making the initial setup invalid. By treating individual ice shelves, it would also give greater detail in the coupling with ocean-model results. Furthermore, it is not complicated to implement this in a dynamical fashion.

2. While details on the implementation in PISM are given, the presented material doesn't go further than applying it to the BEDMAP2 geometry (at a given spatial resolution). Basically, the link with PISM is non-existent. It would therefore be appropriate to see how the model behaves when really applied to PISM, i.e., for an

initial state (for instance spinup) close to the present day, where ice shelves are actually evolving. The only experiments shown are diagnostic, but a prognostic run would really demonstrate the capacity of the PICO coupler. Furthermore, a short run forward in time would reveal how sub-shelf conditions adapt to changing grounding-line position.

**Detailed comments**

- P2, L1: A reference to Thoma et al (2008) 'Modelling Circumpolar Deep Water intrusions on the Amundsen Sea continental shelf, Antarctica - GRL' would also be in place.

- P2, footnote: http://www...

- P5, L20: form stress

- P5, Eq (4): I guess there is an error in this equation, since the value of $\rho$ will be a fractional value and not a local density. In fact, the original equation from OH10 reads

$$\rho = \rho_* \left(1 - \alpha(T - T_*) + \beta(S - S_*)\right)$$

where $T_*$ = 0C and $S_*$ = 34 PSU. In combination with Eq (3), this then leads automatically to Eq (A9), where these two *-values are cancelled out.

- P6, L3: Neglecting heat flux into the ice, ...

- P6, L19: see major remark: why not using ice shelves instead of basins?

- P6; 2.3: What happens if the shelf is really thin or absent? Is the box model still applied for these contacts with the ocean?

[Figure]

- P7, 2.4: This section is irrelevant as long as the behaviour in PISM is not shown. I wold therefore like to see such a simulation with the coupled system that evaluates the coupler for ice-sheet modelling beyond the diagnostic case. It would also be useful to see how it behaves when the grounding line retreats.

- P8, L4: Even when the average of grid cells of the adjacent box is used to connect with the next box, a sharp transition (maybe less sharp) exists, as can been seen from Fig. 5. I agree that the box model can be applied locally to the shelf geometry through local variation of the ice pressure $p$ that changes the local temperatures and salinities. However, the boundary condition to a box is given by the conditions of the adjacent box, not the conditions of a series of elements that are closest (physically this makes no sense). So, why not using just the adjacent box properties (which are mean values anyway) as a boundary condition to the local values with a box? Wouldn't this also make the model more conservative (see discussion on P15)?

- P8, 3: See remark on basins versus ice shelves.

- P11, L1-4: I would not consider criterion 3 and 4 criteria. 1 and 2 definitely are the basis of the overturning model; 3 and 4 are limits obtained from tuning (or validation with respect to two ice shelves), not criteria.

- P13, L2: See previous remark. I would't state that generally the melt rates are highest in the vicinity of the grounding line; that is an assumption made by the model and should be stated as such.

- P13, L5: Ice shelf thickness, hence pressure $p$ is a factor that has a relatively large impact on melt rates. This is also the reason why highest melt rates within box 1 are found nearest to the grounding line. To me this is a more important observation.

- P15, Discussion: Some discussion on the limits of the model should be given. Coupled ocean-ice sheet/shelf models show that not always the maximum melt is reached at the grounding line (e.g., De Rydt and Gudmundsson (2016) Coupled ice shelf-ocean modeling and complex grounding line retreat from a seabed ridge, JGR) . Also, what are the consequences of considering overturning circulation for all ice shelves; the assumption of always having melt in box 1 and decreased melt/accretion towards the front, stable vertical stratification, and relating ocean temperature (not circulation or intrusion of CDW underneath ice shelves) to sub-shelf melt?

- P17, L2: This has not been shown in this paper and remains 'potential'. Although I recognize the potential of it.

---

## Referee Comment (RC2) · Anonymous Referee #2 · 4 Aug 2017

**Summary:**

This manuscript describes the Potsdam Ice-shelf Cavity mOdel (PICO), a new model of Antarctic ice shelf cavity circulation and the heat and freshwater exchange between the ocean and ice shelf.  PICO simulates the two-dimensional overturning circulation within Antarctica's ice shelf cavities that is driven by the 'ice-pump mechanism' described by the authors on page 3 lines 7-8 as "melting at the ice shelf base near the grounding line reduces salinity and the ambient ocean water becomes buoyant, rising along the ice shelf base towards the calving front."  The model consists of a number of connected boxes.  Denser waters from the continental shelf are transported unmodified to the grounding line in a single box, Box 0.  The outflow of buoyant waters beneath the ice shelf occurs within a series of adjoining connected boxes (Box 1 through Box n) that span the area between at the grounding line and the ice shelf calving front.  T and S properties upper layer outflow boxes become progressively modified following ice melting and refreezing.

The key assumptions in this model are:
1) the inflow volume flux, q, is proportional to the density difference between the relatively denser deeper waters of the shelf outside the ice shelf (Box 0 or $B_0$) and the relatively lighter waters near the grounding line (Box 1 or $B_1$).
2) T and S on the outside the cavity do not evolve
3) ice shelf cavity circulation is steady-state
4) no diffusive exchanges of T and S in the vertical and horizontal directions
5) turbulent exchange parameters are constant (no flow rate dependence)
6) salinity at the ice-ocean boundary layer is that of the far-field
7) no conductive heat fluxes from the ocean into the ice shelf
8) no contribution of ice shelf meltwater into the volume flux through the upper layer boxes
9) the ocean equation of state is linearized
10) the prescribed inflow boundary conditions for all boxes of $B_k$, k > 0 is set as the mean of the ice-model grid cell boxes in $B_{k-1}$ that are adjacent to $B_k$

Each upper layer PICO grid cell maps to many ice shelf model grid cells.  In each PICO box the ocean-ice heat and freshwater fluxes are calculated separately.

The two principal unknowns for the model are (1) the constant of proportionality, C, that sets the strength of the density-driven inflow and (2) the turbulent heat flux coefficient, $gamma_T$*.

The authors use PICO with modern day values of T and S around the continental shelf to determine which set of C and $gamma_T$* yield the best fit to modern day ice shelf melt rates.  Using those parameter values for the entire domain they then calculate the melt rate response to varying ocean temperature variations by +/-2 C for Pine Island, Filchner-Ronne, and all Antarctica ice shelves.  Melting in the cold Filchner-Ronne and Antarctica as a whole responds

approximately quadratically with increasing T.  In contrast, melting of the warm PIG increases approximately linearly.  Both melt rate responses are consistent with earlier modelling results.

**Major Comments:**

I found this paper to be clear and well written.  While I could not follow the reasoning for several of the model's assumptions I appreciated that the assumptions were articulated.

PICO is a simplified version of the Olbers and Hellmer 2010 model (OH10).  If there is any major criticism to be made about this work it is that it was not clear to me why deriving a model that could be "analytically solved" is so important.  While there is of course an argument for using the simplest useful model, I found myself wondering whether bits of the physics were being tossed out (e.g., heat conduction into the ice, neglect of contribution of meltwater into the volume transport, use of far-field salinity instead of the boundary-layer salinity one expects from the three-equation model, neglect of velocity dependence on turbulent flux parameter) just to make the system linear.

If PICO's simplifications assumptions were indeed made to yield a linear system of equations that could be directly solved for the purpose of numerical expediency (it is qualitatively described as "very fast" Page 16, Line 13) then it would have been useful for the authors to somehow demonstrate that or at least estimate the computational cost savings enjoyed in comparison to a more complex model such as OH10.  At the end of the day, having a nonlinear set of equations that conserves energy and mass but must be solved iteratively might be preferable to one that doesn't conserve either.

Finally, if ruthless simplification is the goal then the authors have could have gone one step further.  In its present form the ocean inflow and outflow of PICO has no lateral dependence (no effect of Earth's rotation through the Coriolis effect).  Instead of dividing the ice shelves into concentric rings and solving the equations in each ice-model grid cell within each box, the authors could have collapsed each ice shelf into just two dimensions and solved the equations not on the ice model grid but only in the box domain.  The melt rates could then be imposed back onto the 3D ice shelf.  The reason I mention this is that the concentric ring approach taken by PICO yields very strange patterns in ice melt rates (see annotated red arrows on my excerpt of Fig 5 below).  Imposing that pattern of into the ice shelf model would almost certainly lead to an undesirable outcome in the long run.

[Figure]

Filchner-Ronne
0.12 m/a
0.28-0.49 m/a

**Minor Comments:**

Some comparison of spatial patterns of inferred ice shelf melt rates from observations would be helpful, especially for some of the larger ice shelves, would be helpful. A zoom-in on the spatial pattern of some of the faster melting ice shelves like PIG is also advised.

Given uncertainties in the ice shelf temperatures and the ice shelf melt rates used to fit of the two free model parameters, I'd caution the authors against putting too much emphasis on the nominal values of the parameters.

Page 9, Line 5: "sieve criteria?" This is not a common term.

Page 6, lines 9-15 should be clarified.

Page 12, line 6: I don't find any discussion about the procedure to determine the best-fit parameters. Perhaps you it was included in one point but it seems to be missing now. Page 12, Line 6 refers to the best fit values "found in Sect. 3.1" but 3.1 just describes the criteria and the parameter space.

I think the conclusions section could be rewritten.

Page 16, Line 25: your model also does not "fully reflect the circulation below ice shelves".

Page 16, Line 26: you didn't validate your model to present-day ocean conditions and ice geometries, you found a set of free model parameters that yields best fit to modern day ice shelf melt rates.

Page 16, Line 17: I would not say that PICO "accurately" reproduces the "general" pattern of ice shelf melt with higher melting at the grounding line etc. I'd say that PICO qualitatively reproduces the general pattern of ice shelf melt with higher melting at the grounding line etc.

Page 16, Line 30: I'd back off on the claim that you found two calibrated parameters that are "valid for the whole ice sheet". You found a set of parameters that best fit (using a method that was not described) present-day ice shelf melt rates.

---

## Author Comment (AC1) · 15 Nov 2017

Please find our response to all reviewer comments as well as a new supplementary video showing a transient simulation with PISM+PICO in the attached folder. The changes in the manuscript (generated by latexdiff) can be found at the end of the PDF file.

Please also note the supplement to this comment:
https://www.the-cryosphere-discuss.net/tc-2017-70/tc-2017-70-AC1-supplement.zip

---

## Author Response (AR1)

**Response to Reviewer Comments**
**Date: 20 September 2017**
**By R. Reese, T. Albrecht, M. Mengel, X. Asay-Davis and R. Winkelmann**

Journal: TC

Title: Antarctic sub-shelf melt rates via PICO

Author(s): R. Reese, T. Albrecht, M. Mengel, X. Asay-Davis and R. Winkelmann

MS No.: tc-2017-70

MS Type: Research article

First of all, we would like to thank the editor Eric Larour, the reviewer Frank Pattyn and the second, anonymous reviewer for their helpful and excellent comments and their efforts to create the detailed reviews! In our revision of the manuscript we addressed the main issues:

1. We rewrote the discussion and conclusion.

2. We changed the model to solve the governing equations per ice shelf (instead of per basin) and updated all figures and tables correspondingly.

3. We performed a forward simulation to show the capabilities of the model in combination with PISM.

4. We added a comparison with observed melt rate patterns for Filchner-Ronne and Ross ice shelves.

We provide detailed answers to all comments below. The reviewer's comments are given in black and the authors comments in blue. The changes made to the main document can be found at the end (created with latexdiff). Page and line numbers given below relate to this document.

**Referee 1: F. Pattyn**

**General comments**
This is a great paper that fills in the gap that is currently existing in linking large scale ocean to ice-sheet models. At this time, it is probably the best alternative to fully coupled ice-shelf - ocean cavity circulation in order to determine basal melt rates underneath ice shelves. The method is based on the Olbers and Hellmer box model (OH10), but extended to two plan-view dimensions. While it encompasses a series of approximation to this simple model, it is superior to current parametrizations used in large-scale ice-sheet modelling relating melt rates to ice draft. The paper is well written and gives sufficient details on how the model is derived from OH10 and implemented numerically.

The basic premise of the PICO ocean-coupler (if I may call it so) is that circulation in ice shelf cavities is based on vertical overturning. Parameters are therefore chosen in such a way that overturning is applied to all sub-shelf cavities around Antarctica, leading to sub-shelf melting close to the grounding line and accretion (if conditions apply) away from it. Results of the model applied to present-day Antarctic ice shelves gives sub-shelf melt rates close in agreement with observed values. A brief sensitivity analysis demonstrates the effect of ocean temperatures on sub-shelf melt rates.

While the model is definitely interesting for use as an ocean coupler (in absence of fully-coupled solutions), some care should be taken in its future use: it is based on stable vertical stratification, it only considers overturning circulation under ice shelves, it neglects Coriolis effects, and it relates ocean temperature (not circulation or intrusion of CDW underneath ice shelves) to sub-shelf melt. However, major advantages are that it considers the physics of the overturning circulation and that ice shelf size (given by the number of sub-shelf boxes) and distance to grounding line and ice shelf front matters.

We would like to thank Frank Pattyn very much for his enthusiastic evaluation of our manuscript and appreciate his helpful comments for improving our study.

I have only two major comments on the paper:

1. Why using basins and not individual ice shelves to link to mean values of $T_0$ and $S_0$ ? It seems to me that ocean circulation (and temperature/salinity) is related to individual ice shelves and not to drainage basins, which are governed by inland ice flow. Furthermore, during prognostic simulations, these drainage boundaries may change over time, making the initial setup invalid. By treating individual ice shelves, it would also give greater detail in the coupling with ocean-model results. Furthermore, it is not complicated to implement this in a dynamical fashion.

   Thanks for bringing this important point! We changed the model as suggested: we now identify single ice shelves and solve the box model equations individually for each ice shelf and not basin-wide as done before. We updated all figures and tables and changed the text to be consistent with the model improvement, e.g. in Sect. 2.3 on page 6 of the latex-diff manuscript attached below. The effect of our update is negligible for the melt rates of large ice-shelves; smaller ice shelves show slightly higher melt rates, compare Figure 1 below.

   The input of PICO - which itself is a simple model - should predominantly represent the large-scale characteristics of the ocean surrounding the ice shelf. Neglecting ocean dynamics, we here approximate water masses from "source" regions upstream of the ice shelf cavities by averaging ocean properties on the depth of the continental shelf within a basin (shown in Fig. 2 of the main manuscript). These 19 regions capture the large-scale characteristics of the water masses surrounding the Antarctic continent (compare

[Figure]

**Figure 1:** Comparison of sub-shelf melt rates underneath ice shelves in the Amundsen Sea region with the previous method (a) and the updated method that solves the box model per ice shelf (b).

Fig. 1 in Schmidtko et al. (2014)). We now link ocean input in box $B_0$ for each ice shelf individually such that the ocean input adjusts dynamically when an ice shelf evolves across basin boundaries: The temperature and salinity values of all basins that the ice shelf belongs to are averaged, weighted by the fractional area of the shelf in the respective basin. We changed the description of PICO input in the main document accordingly, e.g., in Sect. 3, page 9, line 1ff in the latex-diff document attached.

2. While details on the implementation in PISM are given, the presented material doesn't go further than applying it to the BEDMAP2 geometry (at a given spatial resolution). Basically, the link with PISM is non-existent. It would therefore be appropriate to see how the model behaves when really applied to PISM, i.e., for an initial state (for instance spinup) close to the present day, where ice shelves are actually evolving. The only experiments shown are diagnostic, but a prognostic run would really demonstrate the capacity of the PICO coupler. Furthermore, a short run forward in time would reveal how sub-shelf conditions adapt to changing grounding-line position.

This is indeed a good point. We now provide a forward simulation based on an equilibrium state of the Antarctic Ice Sheet with a video in the Supplementary Material and discuss the adaptation of PICO to grounding line movement in the newly added Section 3.3. Starting from equilibrium conditions for Antarctica, ocean temperatures increase linearly over 50 years until an ocean-wide warming of $1°C$ is reached and are held constant from then on. The movie shows the temporal evolution of the ocean temperature input and the sub-shelf melt rates for the ice shelf adjacent to Pine Island Glacier as well as the Filchner-Ronne Ice Shelf. The melt rates in both ice shelves increase, especially in the first boxes (spatial distribution for FRIS in the upper right panel and for PIG in the lower right panel). Subsequent ice-shelf thinning reduces the buttressing and the grounding lines retreat with the ocean boxes and melt rates adjusting accordingly.

**Detailed comments**

- P2, L1: A reference to Thoma et al (2008) "Modelling Circumpolar Deep Water intrusions on the Amundsen Sea continental shelf, Antarctica - GRL" would also be in place.
  Done.

- P2, footnote: http://www...
  Done.

- P5, L20: form stress
  Done.

- P5, Eq (4): I guess there is an error in this equation, since the value of $\rho$ will be a fractional value and not a local density. In fact, the original equation from OH10 reads

$$\rho = \rho_*(1 - \alpha(T - T_*) + \beta(S - S_*))$$

  where $T_* = 0C$ and $S_* = 34$ PSU. In combination with Eq (3), this then leads automatically to Eq (A9), where these two $*$-values are cancelled out.
  Thanks for pointing this out, this is absolutely true. We corrected the formula.

- P6, L3: Neglecting heat flux into the ice, ...
  Done.

- P6, L19: see major remark: why not using ice shelves instead of basins?
  Good point! We changed the code and the text here accordingly, see Sect. 2.3.

- P6; 2.3: What happens if the shelf is really thin or absent? Is the box model still applied for these contacts with the ocean?
  If the ice-shelf is really thin, the model is still applied. In this case, melt rates would be low or even negative (representing accretion), depending on the input oceanic temperatures. The model is not applied in the absence of an ice shelf, i.e., melting along vertical ice cliffs - as for example at the termini of some Greenland outlet fjords - is not modeled by PICO. We added this to the main text, see page 7, line 18 of the latex-diff attached.

- P7, 2.4: This section is irrelevant as long as the behaviour in PISM is not shown. I would therefore like to see such a simulation with the coupled system that evaluates the coupler for ice-sheet modelling beyond the diagnostic case. It would also be useful to see how it behaves when the grounding line retreats.
  Thanks for bringing this to our attention. To demonstrate the behaviour of the coupled system, we made a forward run with PICO, as described in the newly added Sect. 3.3. The adjustment of the melt rates to grounding line migration can be seen in our movie (added to the Supplementary Information) with a detailed explanation in the answer to your second major comment.

- P8, L4: Even when the average of grid cells of the adjacent box is used to connect with the next box, a sharp transition (maybe less sharp) exists, as can been seen from Fig. 5. I agree that the box model can be applied locally to the shelf geometry through local variation of the ice pressure p that changes the local temperatures and salinities. However, the boundary condition to a box is given by the conditions of the adjacent box, not the conditions of a series of elements that are closest (physically this makes no sense). So, why not using just the adjacent box properties (which are mean values anyway) as a boundary condition to the local values with a box? Wouldn't this also make the model more conservative (see discussion on P15)?

  Thanks for bringing this great suggestion. We changed the code accordingly and updated all tables and figures as well as the text (e.g. in Sect. 2.4, page 8, lines 15ff) correspondingly. This did substantially improve the deviation of incoming and outgoing heat fluxes from $-867.19\ GJ\,s^{-1}$ (which is equivalent to $< 5\%$ of the latent heat flux for melting) to $403.63\ GJ\,s^{-1}$ (which is eqivalent to $< 2.5\%$ of the latent heat flux for melting). We do not achieve perfect conservation of energy here, since temperature in Box $B_1$ is a non-linear function of pressure (see Eqn. A12): Averaging the results for all ice-model grid cells does not equal the result for one pressure value averaged over all ice shelf grid cells in Box $B_1$ (see also discussion on page 16, lines 7ff).

- P8, 3: See remark on basins versus ice shelves.

  We re-formulated this part of the text consistent with the updated model code (page 9 lines 1ff). Melt rates are now calculated for each shelf individually, see also our answer to your major comment #1.

- P11, L1-4: I would not consider criterion 3 and 4 criteria. 1 and 2 definitely are the basis of the overturning model; 3 and 4 are limits obtained from tuning (or validation with respect to two ice shelves), not criteria.

  Thanks for this remark, this is absolutely correct. We renamed criteria 3 and 4 to "observational constraints" (1) and (2) in order to distinguish these from the qualitative model criteria (1) and (2), see page 12 lines 15ff and distinguish the criteria (black contour line) from the observational constraints (green contour line) in Fig. 4.

- P13, L2: See previous remark. I would't state that generally the melt rates are highest in the vicinity of the grounding line; that is an assumption made by the model and should be stated as such.

  We changed the text to make this clear: " Consistent with the model assumptions, melt rates ...".

- P13, L5: Ice shelf thickness, hence pressure p is a factor that has a relatively large impact on melt rates. This is also the reason why highest melt rates within box 1 are found nearest to the grounding line. To me this is a more important observation.

Thanks for bringing these points to discussion! We added this observation. The text now reads:

"The melt pattern depends on the local pressure melting temperature. Thus, melt rates are highest where the shelf is thickest, *i.e.,* near the grounding lines within box $B_1$. Furthermore, freezing can occur for relatively thin ice in the same box in which melting occurs where the ice shelf is thicker. "

- P15, Discussion: Some discussion on the limits of the model should be given. Coupled ocean-ice sheet/shelf models show that not always the maximum melt is reached at the grounding line (e.g., De Rydt and Gudmundsson (2016) Coupled ice shelf-ocean modelling and complex grounding line retreat from a seabed ridge, JGR) . Also, what are the consequences of considering overturning circulation for all ice shelves; the assumption of always having melt in box 1 and decreased melt/accretion towards the front, stable vertical stratification, and relating ocean temperature (not circulation or intrusion of CDW underneath ice shelves) to sub-shelf melt?

Thanks for bringing up these important points! We incorporated these into the Discussion:

**Location of maximum melting**

PICO's design does not allow for capturing melt patterns on very local scales as described by De Rydt and Gudmundsson (2016). Their finding is consistent with further ocean simulations and observations which show lower melt rates near grounding lines but to our knowledge this is not yet widely accepted as a robust phenomenon, nor are the length or depth scales over which melt rates decay toward grounding lines well established in the literature. It is particularly important and an open question, if and how much ice dynamics are affected when the same melt flux is distributed over an area near the grounding line or a smaller area not quite reaching the grounding line. Here, we aim at covering the larger-scale features of basal melt, and in that sense we think that PICO does a good job at covering the melt in the region close to the grounding line (in PICO box $B_1$, see Tables 1 and 2 and the newly added Fig.S4 which compares PICO melt rates to observations).

We added a discussion of this limitation to the manuscript, see page 17 lines 25ff and 28ff.

**Overturning**

Following your suggestions, overturning is now calculated individually for each ice shelf (instead of per basin), yielding slightly higher melt rates for small ice shelves (see also the response to your major comment 1). The simplifying assumptions of PICO imply that one overturning value is determined per shelf; the local pattern of ocean currents underneath the ice shelf is not resolved in detail. We stated this in the discussion on page 17 in lines 14ff of the latex-diff document attached.

**Positive melt rates in box $B_1$**

Thanks for mentioning this, we made sure that we state this model assumption in discussion on page 18 in lines 1f of the latex diff manuscript. Melting in box $B_1$ is a necessary

condition for the box model, but also for the ice pump. Observations indicate that melt rates are almost universally positive close to the grounding line and small areas of freezing near the grounding line would be averaged with larger areas of melting in box $B_1$ of our model, see for example the observed sub-shelf melt rates in Fig. S4 with average melt rates annotated in each box.

**Stable vertical stratification**

Thanks for bringing this important point, we added it in the discussion on page 18 in lines 8-14: PICO builds on the findings of OH10 that the ocean column beneath the ice shelf is in general stratified when a steady state is reached. Furthermore, the overturning circulation as formulated in the box model is prevented from reaching neutral density and detaching from the ice-shelf base while flowing towards the shelf front. In this case, the spatial pattern of melting closer to the calving front of cold ice shelves is not represented well which may explain why PICO melt rates averaged over boxes located towards the calving front are negative in FRIS and Ross while this is not necessarily the case in the observations (see Tables 1 and 2 in this document).

**CDW and ocean dynamics**

PICO input is determined by averaging bottom temperatures and salinities over the continental shelf, this is done for 19 different basins. This means that PICO - which itself is a coarse model - will miss the nuances of how ocean currents transport and modify CDW over the regions being averaged. The procedure to determines melt rates in PICO is based on the assumption that ocean water that is present on the continental shelf can access the ice shelf cavities and reach their grounding lines. This implies for example, that barriers like sills that may prevent intrusion of warm CDW are not accounted for and might explain why PICO melting is too high for the ice shelves located along the Southern Antarctic Peninsula. Any such phenomena could be tested by varying the ocean input of PICO. We added this limitation to the discussion on page 18 in lines 15ff.

- P17, L2: This has not been shown in this paper and remains "potential". Although I recognize the potential of it.

  Thanks for pointing this out - we added the forward simulation to demonstrate PICO's capabilities to adapt to grounding line migration.

**Anonymous referee 2**

The comment was uploaded in the form of a supplement:

https://www.the-cryosphere-discuss.net/tc-2017-70/tc-2017-70-RC2-supplement.pdf

**Summary**

This manuscript describes the Potsdam Ice-shelf Cavity mOdel (PICO), a new model of Antarctic ice shelf cavity circulation and the heat and freshwater exchange between the ocean and ice shelf. PICO simulates the two-dimensional overturning circulation within Antarctica's ice shelf cavities that is driven by the "ice-pump mechanism" described by the authors on page 3 lines 7-8 as "melting at the ice shelf base near the grounding line reduces salinity and the ambient ocean water becomes buoyant, rising along the ice shelf base towards the calving front." The model consists of a number of connected boxes. Denser waters from the continental shelf are transported unmodified to the grounding line in a single box, Box 0. The outflow of buoyant waters beneath the ice shelf occurs within a series of adjoining connected boxes (Box 1 through Box n) that span the area between at the grounding line and the ice shelf calving front. T and S properties upper layer outflow boxes become progressively modified following ice melting and refreezing.

The key assumptions in this model are:

1. the inflow volume flux, q, is proportional to the density difference between the relatively denser deeper waters of the shelf outside the ice shelf (Box 0 or $B_0$) and the relatively lighter waters near the grounding line (Box 1 or $B_1$).

2. T and S on the outside the cavity do not evolve

3. ice shelf cavity circulation is steady-state

4. no diffusive exchanges of T and S in the vertical and horizontal directions

5. turbulent exchange parameters are constant (no flow rate dependence)

6. salinity at the ice-ocean boundary layer is that of the far-field

7. no conductive heat fluxes from the ocean into the ice shelf

8. no contribution of ice shelf meltwater into the volume flux through the upper layer boxes

9. the ocean equation of state is linearized

10. the prescribed inflow boundary conditions for all boxes of $B_k$ ,$k > 0$ is set as the mean of the ice-model grid cell boxes in $B_{k-1}$ that are adjacent to $B_k$.

Each upper layer PICO grid cell maps to many ice shelf model grid cells. In each PICO box the ocean-ice heat and freshwater fluxes are calculated separately.

The two principal unknowns for the model are (1) the constant of proportionality, C, that sets the strength of the density-driven inflow and (2) the turbulent heat flux coefficient, gammaT*.

The authors use PICO with modern day values of T and S around the continental shelf to determine which set of C and gammaT* yield the best fit to modern day ice shelf melt rates. Using those parameter values for the entire domain they then calculate the melt rate response to varying ocean temperature variations by +/-2 C for Pine Island, Filchner-Ronne, and all Antarctica ice shelves. Melting in the cold Filchner-Ronne and Antarctica as a whole responds approximately quadratically with increasing T. In contrast, melting of the warm PIG increases approximately linearly. Both melt rate responses are consistent with earlier modelling results.

We would like to thank the reviewer very much for her/his effort, the helpful comments and the great evaluation!

**Major comments**

I found this paper to be clear and well written. While I could not follow the reasoning for several of the model's assumptions I appreciated that the assumptions were articulated.

Thanks for this positive assessment. Based on your comments, we tried to make the assumptions of PICO clearer and added discussion on these and the scope of the model to the main document.

PICO is a simplified version of the Olbers and Hellmer 2010 model (OH10). If there is any major criticism to be made about this work it is that it was not clear to me why deriving a model that could be "analytically solved" is so important. While there is of course an argument for using the simplest useful model, I found myself wondering whether bits of the physics were being tossed out (e.g., heat conduction into the ice, neglect of contribution of meltwater into the volume transport, use of far-field salinity instead of the boundary-layer salinity one expects from the three-equation model, neglect of velocity dependence on turbulent flux parameter) just to make the system linear.

Thanks for bringing this to our attention! We added more discussion on the choices made in PICO and the overall scope of the model.
The aim of PICO is to introduce a simple model that computes physically-based sub-shelf melt rates and is hence preferred to parametrizations often used in current large-scale ice-sheet modelling studies that relate melt rates to the depth of the ice shelf base. We designed PICO in a way that it is easy to implement, avoiding iterative techniques or other non-linear solvers that might complicate the numerics. Nevertheless, we agree with the reviewer that all approximations

made need to be physically justified. We tried to do so here and also tried to improve the main text in this sense.

We base our discussion here on your nice summary of PICO's assumptions above. Some of these assumptions are PICO-specific, some stem from the OH10 model and others are generally made in ocean modelling. We discuss all these and their implications here and added this in the main text correspondingly, e.g., on page 4 in lines 7ff of the latex-diff document attached.

There may be a misunderstanding about assumption 2 ('T and S on the outside the cavity do not evolve'): The evolution of the input of $T$ and $S$ in PICO is possible. The model accepts time-dependent ocean forcing with an example given in the new Supplementary video which is discussed in the newly added Sect 3.3. The input T and S could also come from an evolving ocean model that includes freshwater fluxes taken from PICO.

Assumption 9 ('the ocean equation of state is linearized') is widely used in high-resolution models for ocean dynamics. Assumptions that are taken on from the OH10 model are:

1. the inflow volume flux, q, is proportional to the density difference between the relatively denser deeper waters of the shelf outside the ice shelf (Box 0 or $B_0$) and the relatively lighter waters near the grounding line (Box 1 or $B_1$).

5. turbulent exchange parameters are constant (no flow rate dependence)

7. no conductive heat fluxes from the ocean into the ice shelf

8. no contribution of ice shelf meltwater into the volume flux through the upper layer boxes

We revised the text to make this clearer, e.g., on page 4 lines 7ff and by sorting the discussion correspondingly, see pages 17 and 18.

Neglecting the velocity dependence of the turbulent flux parameters was inherited from the OH10 model. Holland and Jenkins (1999) estimate that including constant vertical heat advection reduces the computed melt rates by about 10% (their Fig. 7c). We hence omit this in PICO, but we agree that heat conduction would be a natural term to include in a coupled ice-ocean model and hence we envisage to do so in the next model version. We estimated the contribution of melt water to the volume flux to be small ($< 1\%$ of the total overturning transport for the entire continent) and hence we regard its neglect as justified. PICO-specific assumptions are:

3. ice shelf cavity circulation is steady-state

4. no diffusive exchanges of T and S in the vertical and horizontal directions

6. salinity at the ice-ocean boundary layer is that of the far-field

The main assumption for PICO is that the overturning circulation in the cavity is in steady-state (3.). Since PICO's aim is to provide sub-shelf melt rates for large-scale ice sheet simulations and because of the much longer time scales of ice dynamics in comparison to ocean dynamics, we regard the assumption as justified. Using this assumption facilitates the adaptive box

adjustment in the case of grounding line migration or calving, especially since PICO transfers the box model approach into two horizontal dimensions. In the non-steady OH10 model, box extents and locations are fixed and to allow for evolving boxes, volume fluxes arising from box movement would add additional terms to the models' transport equations.

OH10 state that diffusive transport is small when their model reaches steady state, which justifies assumption 4, see also page 4 lines 8-10. We emphasize that using the far-field salinity (assumption 6) is only used for the computation of the sub-shelf melting directly. This linearisation of the melt equation was proposed for example by (McPhee, 1992) and was found to yield realistic heat fluxes (see e.g. Holland and Jenkins, 1999; McPhee, 1999). As further discussed in the response to your comment regarding Page 6, lines 9-15, we do not apply this for the further calculation of the temperatures and salinities of the boxes, respectively (this would not be supported by the model equations, see discussion added on page 20, lines 25ff). While we plan to address this in the next model version, we hence believe the use of the 2-equations model for the calculation of melt rates is justified here.

Based on the reviewers' comments, we changed the way box-box transitions in PICO are done. Assumption 10 ('the prescribed inflow boundary conditions for all boxes of $B_k$ ,$k > 0$ is set as the mean of the ice-model grid cell boxes in $B_{k-1}$ that are adjacent to $B_k$') reads now

10B. the prescribed inflow boundary conditions for all boxes of $B_k$ ,$k > 0$ is set as the mean of the ice-model grid cell boxes in $B_{k-1}$

The advantage and scope of the model described in this paper is that it is easy to implement and to verify. We hope that the additions to the text make the underlying reasoning of our approach transparent.

If PICO's simplifications assumptions were indeed made to yield a linear system of equations that could be directly solved for the purpose of numerical expediency (it is qualitatively described as "very fast" Page 16, Line 13) then it would have been useful for the authors to somehow demonstrate that or at least estimate the computational cost savings enjoyed in comparison to a more complex model such as OH10. At the end of the day, having a nonlinear set of equations that conserves energy and mass but must be solved iteratively might be preferable to one that doesn't conserve either.

Thanks for stating this. We changed the text to make the scope of our model design clearer, e.g., on page 17 in lines 10-15 of the latex-diff below. We agree that solving the entire system would not be much more expensive - given the time needed to solve the ice model equations (especially the SSA).
But, having these simple model equations has some advantages: We designed the model in a way that is easy to implement, such that other ice-sheet models can easily adopt a similar approach. Moreover, given the analytic solution, the numerical solution of PICO can easily be checked

manually, e.g., for an ice shelf with a constant thickness.

Furthermore, we argue that the errors introduced by the approximations and simplifications made in the model are of second order (see the answers to your comments above).

The mass loss in the model arises mainly from the assumption that the overturning is constant along the boxes, an assumption that introduces small errors and is similarly done in the OH10 model.

There seems to be a misunderstanding concerning the linearity of the system of equations. The temperature in box $B_1$ is a non-linear function of pressure. Since we locally adjust the sub-shelf melt rates to the pressure melting point and use the box-wide temperature average as the boundary condition for the next box, PICO is not perfectly conserving. We find the resulting error to be small (less than 2.5% of the latent heat flux due to melting). Conservation of energy could be obtained by following your idea below and the original method of OH10, solving for a single sub-shelf melt rate with piecewise constant T and S properties in each box. The disadvantage of such an approach would be that the transition between the melt rates of the boxes would be much sharper. Also, any effect of the variation of the pressure melting point between elements within a box would be lost.

Finally, if ruthless simplification is the goal then the authors have could have gone one step further. In its present form the ocean inflow and outflow of PICO has no lateral dependence (no effect of Earth's rotation through the Coriolis effect). Instead of dividing the ice shelves into concentric rings and solving the equations in each ice-model grid cell within each box, the authors could have collapsed each ice shelf into just two dimensions and solved the equations not on the ice model grid but only in the box domain. The melt rates could then be imposed back onto the 3D ice shelf. The reason I mention this is that the concentric ring approach taken by PICO yields very strange patterns in ice melt rates (see annotated red arrows on my excerpt of Fig 5 below). Imposing that pattern of into the ice shelf model would almost certainly lead to an undesirable outcome in the long run.

This is a relevant point that is important to discuss. We did not follow the approach mentioned by the reviewer because this would not allow us to take the local pressure in every ice-sheet model grid cell into account. Collapsing the rings and solving per box (as done in the original OH10 model), would hence not allow for spatial patterns but yield a single melt rate per box. Imposing one melt rate for the entire box would yield much sharper transitions of the melt rates between the individual boxes. Also, there would be no local adjustment of the melt rates in areas thinner or thicker than the average depth within a box - suppressing an important negative feedback: If melting thins the ice-shelf, the local pressure melting point at the ice-shelf base increases and melting is reduced.

From the approach outlined in the paper, spatial melt rate patterns arise in the diagnostic solution discussed in the paper and based on the Bedmap2 geometry. In order to test your

concerns, we did a forward simulation with PICO over 300 years. The resulting evolution of the melt rates of the ice shelves in the Amundsen region and Filchner-Ronne Ice Shelf and the adjustment to grounding line migration is displayed in the newly added Supplementary Video S1. In this forward run, we find that the melt rate patterns do not yield the problems you stated. As you can see in the prognostic simulation, the 'patches' do not show on the ice-shelf sub-surface after 300 years (Figure 3 in this document). The negative pressure feedback tends to reduce them. The boxes' borders are slightly visible in Filchner-Ronne Ice Shelf (left panel), which we suspect to be much sharper if one melt rate per box was determined.

[Figure]

**Figure 2:** Inserted by reviewer 2.

[Figure]

**Figure 3:** Ice shelf sub-surface of Filchner Ronne Ice Shelf (left panel) and the ice shelves in the Amundsen region (right panel) after 300 years, compare also the video in the Supplementary Information.

**Minor comments**

Some comparison of spatial patterns of inferred ice shelf melt rates from observations would be helpful, especially for some of the larger ice shelves, would be helpful. A zoom-in on the spatial pattern of some of the faster melting ice shelves like PIG is also advised.

This is a good point! We increased the zoom into the Amundsen region with melt rates shown for different model parameters in the Fig. S1 of the Supplementary Informations (also included in the latexdiff below). This region is also displayed in the newly added movie showing a prognostic simulation of sub-shelf melting (Supplementary Video).
We did a comparison with melt rate patterns calculated from observations described in (Moholdt et al., 2014) and obtained from (Moholdt et al., 2016) for the Filchner-Ronne and Ross ice shelves, see the newly added Fig.S4. While the general pattern of melt rates for FRIS and Ross is fairly well represented (missing, e.g., the seasonal intrusion of warm water from the calving front), the melt rates are "smoothed out" over the ice shelf: The locally observed melt rates show larger deviations from the average melting than the melt rates modelled with PICO (compare Fig. S4). Nevertheless, box-wide averages of melting show reasonable agreement from PICO and from the observations (per-box averages are given in Tables 1 and 2 with the boxes shown on top of the observational melt rates in Fig. S4). The PICO melt rates for boxes located towards the calving front are generally negative in FRIS and Ross while this is not necessarily the case in the observations - this might be explained by the fact that the overturning circulation as formulated in the box model is prevented from reaching neutral density and detaching from the ice-shelf base while flowing towards the shelf front.

**Table 1:** Melt rates from the reference simulation as displayed in Fig. S4 compared with observed values from (Moholdt et al., 2014). Melt rates are averaged over the respective box or the entire ice shelf (last row) and given in meter per year.

| FRIS | $m_{observed}$ | $m_{PICO}$ |
|------|---------------|-----------|
|      | $m\,a^{-1}$   | $m\,a^{-1}$ |
| $B_1$ | 0.42 | 0.47 |
| $B_2$ | 0.19 | -0.03 |
| $B_3$ | 0.20 | -0.18 |
| $B_4$ | -0.26 | -0.13 |
| $B_5$ | 0.53 | -0.07 |
| shelf | 0.26 | 0.06 |

**Table 2:** Melt rates from the reference simulation as displayed in Fig. S4 compared with observed values from (Moholdt et al., 2014).

| Ross | $m_{observed}$ | $m_{PICO}$ |
|------|---------------|-----------|
|      | $m\,a^{-1}$   | $m\,a^{-1}$ |
| $B_1$ | 0.05 | 0.47 |
| $B_2$ | -0.01 | -0.03 |
| $B_3$ | -0.01 | -0.18 |
| $B_4$ | 0.02 | -0.13 |
| $B_5$ | 0.28 | -0.07 |
| shelf | 0.08 | 0.06 |

Given uncertainties in the ice shelf temperatures and the ice shelf melt rates used to fit of the two free model parameters, I'd caution the authors against putting too much emphasis on the nominal values of the parameters.

We agree that this is an important point - as Fig. 4 in the main document shows, there is a whole set of parameters such that model criteria (1) and (2) are satisfied and the basin averages of the melt rates agree to some extent with observations. We changed the text accordingly, see page 19 in line 5 of the latex-diff document.

- Page 9, Line 5: "sieve criteria?" This is not a common term.
  Thanks, these are simply criteria, we removed 'sieve'.

- Page 6, lines 9-15 should be clarified.
  This is an important point in the solution of the system of equations. Using the simplified melt rate formulation (called 2-equations model in (Holland and Jenkins, 1999)) was shown to yield realistic heat fluxes (McPhee, 1992) and is hence applicable in this context. For the solution of the transport equations, the underlying assumption that the boundary layer salinity $S_{bk}$ can be replaced by the salinity of the ambient ocean water, here $S_k$, is however not valid: Doing so would reduce the salinity transport equation to $S_{k-1} = S_k$

and hence the overturning circulation $q = C(\rho_0 - \rho_1)$ would become too weak, since at the low temperatures generally present in the Southern Ocean, this circulation is mainly haline-driven. We added this discussion to the Appendix in lines 25ff on page 20 and changed the text to 'This simplification is used only for melt rates, we nevertheless solve for the boundary layer salinity which is central to the solution of the system of equations as detailed in Appendix A'.

- Page 12, line 6: I don't find any discussion about the procedure to determine the best-fit parameters. Perhaps you it was included in one point but it seems to be missing now. Page 12, Line 6 refers to the best fit values "found in Sect. 3.1" but 3.1 just describes the criteria and the parameter space.

  Thanks for bringing to our attention that we forgot to state this! We added this explanation on page 12 lines 28ff. We determine the parameters such that the root-mean-square deviation of average melt rates for Pine Island Glacier Ice Shelf and Filchner-Ronne Ice Shelf is minimized.

- I think the conclusions section could be rewritten.

  We agree and thanks you for your suggestions that we used to re-write the conclusion.

- Page 16, Line 25: your model also does not "fully reflect the circulation below ice shelves".

  Yes, we agree. We changed this to "which do not account for the circulation below ice shelves".

- Page 16, Line 26: you didn't validate your model to present-day ocean conditions and ice geometries, you found a set of free model parameters that yields best fit to modern day ice shelf melt rates.

  We fully agree. We changed this to 'We find a set of possible parameters for present-day ocean conditions and ice geometries which yield PICO melt rates in agreement with average melt rate observations.'.

- Page 16, Line 17: I would not say that PICO "accurately" reproduces the "general" pattern of ice shelf melt with higher melting at the grounding line etc. I'd say that PICO qualitatively reproduces the general pattern of ice shelf melt with higher melting at the grounding line etc.

  Thanks for bringing up this point, we changed the text accordingly on page 19 in line 7.

- Page 16, Line 30: I'd back off on the claim that you found two calibrated parameters that are "valid for the whole ice sheet". You found a set of parameters that best fit (using a method that was not described) present-day ice shelf melt rates.

Yes, we agree. We meant that two parameters are applied for all ice shelves, and that the parameters are not adjusted for ice shelves or basins individually. We hope this is clearer with the text changed to '...using only two calibrated parameters applied to all ice shelves'. We added an explanation of how we determined the best-fit parameters on page 12 in lines 28ff of the latex-diff document attached.

**References**

[revised manuscript text omitted]

**Figure S.1.** Basal melt rates in the  Filchner-Ronne (upper panels) and Amundsen Sea (lower panels) regions for different parameter combinations of the overturning strength $C$ and the effective turbulent heat transfer coefficient $\gamma_T^*$. Grounded ice regions are shown in grey.

[Figure]

**Figure S.2.** Sensitivity of mean sub-shelf melt rates to the ice-sheet model resolution.

[Figure]

**Figure S.3.** Sensitivity of mean sub-shelf melt rates to the maximum number of boxes of PICO.

[Figure]

**Figure S.4.** Comparison of observed sub-shelf melt rates (upper row) from (Moholdt et al., 2016) with melt rates modelled by PICO (lower row) for Filchner-Ronne (left column) and Ross ice shelves (right column). Black contour lines indicate the PICO ocean boxes with annotations giving the box-wide average melt rates respectively. PICO tends to distribute melting, such that melt rate deviations are at a lower order of magnitude than in the observations. Nevertheless, box-wide averages show reasonable agreement.

**Table S.1.** Results from the reference simulation as displayed in Fig. 5.

| basin | $m_\Sigma$ | $m_{\min}$ | $m_{\max}$ | $q$ | $m_\Sigma/q$ | $Q_{in}$ | $Q_{out}$ | $Q_m$ | $Q_\Delta$ | $Q_\Delta/Q_m$ | $b_1$ |
|---|---|---|---|---|---|---|---|---|---|---|---|
| | $\mathrm{Gt\,a^{-1}}$ | $\mathrm{m\,a^{-1}}$ | $\mathrm{m\,a^{-1}}$ | Sv | % | $\mathrm{TJ\,s^{-1}}$ | $\mathrm{TJ\,s^{-1}}$ | $\mathrm{GJ\,s^{-1}}$ | $\mathrm{GJ\,s^{-1}}$ | % | m |
| Wilkins(*) | 320 | 0.26 | 19.80 | 0.32 | 3.06 | 361.30 | 357.90 | 3382.50 | 19.06 | 0.56 | 272 |
| Pine Island | 61 | 12.39 | 21.01 | 0.17 | 1.11 | 188.51 | 187.87 | 645.56 | 2.68 | 0.41 | 439 |
| Thwaites | 53 | 11.44 | 20.90 | 0.13 | 1.27 | 143.41 | 142.85 | 560.51 | 1.13 | 0.20 | 438 |
| Getz | 112 | 2.48 | 10.78 | 0.23 | 1.48 | 260.10 | 258.90 | 1189.23 | 13.98 | 1.18 | 494 |
| Drygalski | 1 | 0.01 | 3.51 | 0.02 | 0.23 | 17.33 | 17.32 | 12.50 | -0.16 | -1.28 | 293 |
| Cook | 7 | 0.70 | 5.25 | 0.05 | 0.38 | 60.87 | 60.80 | 72.20 | -0.10 | -0.14 | 458 |
| Ninnis | 4 | 1.08 | 6.61 | 0.04 | 0.31 | 48.80 | 48.75 | 47.19 | -0.08 | -0.17 | 514 |
| Mertz | 6 | 0.38 | 4.60 | 0.04 | 0.45 | 43.27 | 43.21 | 60.18 | -0.14 | -0.23 | 309 |
| Totten | 29 | 5.93 | 14.33 | 0.13 | 0.70 | 144.10 | 143.79 | 309.81 | 1.06 | 0.34 | 677 |
| Shackleton | 9 | -0.21 | 2.70 | 0.07 | 0.40 | 77.62 | 77.52 | 96.93 | 0.48 | 0.50 | 270 |
| West | 9 | -0.08 | 5.26 | 0.07 | 0.38 | 78.82 | 78.73 | 93.34 | -0.17 | -0.18 | 428 |
| Amery | 25 | -1.22 | 5.93 | 0.16 | 0.49 | 175.86 | 175.58 | 269.19 | 11.89 | 4.42 | 674 |
| Baudouin | 22 | -0.25 | 2.73 | 0.12 | 0.59 | 128.92 | 128.68 | 234.16 | 7.03 | 3.00 | 325 |
| Fimbul | 19 | -0.25 | 2.97 | 0.10 | 0.57 | 115.85 | 115.64 | 204.02 | 5.79 | 2.84 | 303 |
| Riiser-Larsen | 13 | -0.22 | 1.83 | 0.09 | 0.46 | 94.92 | 94.78 | 136.36 | 3.84 | 2.82 | 273 |
| Brunt(**) | 11 | -0.16 | 2.30 | 0.08 | 0.40 | 93.81 | 93.69 | 117.21 | 3.31 | 2.83 | 250 |
| Filchner-Ronne | 21 | -0.67 | 1.76 | 0.21 | 0.31 | 236.72 | 236.34 | 225.52 | 152.22 | 67.50 | 839 |
| Ross | 25 | -0.24 | 0.62 | 0.17 | 0.44 | 191.38 | 191.01 | 262.62 | 113.98 | 43.40 | 411 |
| Antarctica | 1718 | -1.22 | 26.91 | 8.51 | 0.62 | 9473.42 | 9454.84 | 18183.19 | 403.63 | 2.22 | - |

For each basin, $m_\Sigma$ is the aggregated melt rate over the entire basin, $q$ the overturning flux computed as average over box $B_1$, $m_\Sigma/q$ estimates the error in mass flux introduced by assuming constant overturning; $Q_{in} = T_0 \times q \times c_p \times \rho_w$ is the heat flux from box $B_0$ in box $B_1$, $Q_{out} = T_n \times q \times c_p \times \rho_w$ the flux from the last box $B_n$ adjacent to the shelf front into the ocean, $Q_m = Lm_\Sigma$ the heat flux due to melting of ice, $Q_\Delta = Q_{in} - Q_{out} - Q_m$ the error in the heat balance. $Q_\Delta/Q_m$ is the error in the heat balance relative to the heat flux for melting which results from the non-linearity of the temperature solution in box $B_1$. The average depth of this box is given by $b_1$. (*) includes also Stange, Bach and George VI ice shelves and (**) also Stancomb Ice Shelf.

**Video. S.1.** Based on an Antarctic equilibrium state at 8km resolution comparable to the state submitted to initMIP (Nowicki et al., 2016), PISM+PICO is forced with time-dependent ocean temperature input: Starting from equilibrium conditions, ocean temperatures increase linearly over 50 years until an ocean-wide warming of $1°C$ is reached. It is then held constant over the next 250 model years. The movie shows the temporal evolution of the ocean temperature input for the ice shelf adjacent to Pine Island Glacier as well as the Filchner-Ronne Ice Shelf (upper left panel). The ocean temperature increase enhances the average sub-shelf melting for both ice shelves (lower left panel, shown in logarithmic scale) with the spatial distribution of the melt rates for both ice shelves on the right hand side. During the simulation, the melt rates in both areas increase, especially in the first box (upper right panel shows Filchner-Ronne Ice Shelf and lower right panel the Amundsen region). Ice-shelf thinning reduces buttressing and causes the grounding lines to retreat (for example near Foundation Ice Stream in Filchner-Ronne Ice Shelf) with the ocean boxes adjusting accordingly.

---

## Referee Report (RR1)

**Antarctic sub-shelf melt rates via PICO (Revised)**

Major comments:

After having read the paper several times I keep coming back to the question of how many of the simplifing choices were made to ensure that the resulting box model equations could be analytically solved?  Reducing a complex system to a form amenable to analytical analysis can be a valuable exercise that can yield insights into a system's behavior.  However, in the case of PICO I have a lingering impression that most of its numerous simplications were made to yield an analytically solvable system simply for the sake of having an analytically solvable system.  If some of the simplifying assumptions were lifted and the resulting nonlinear system had to be solved iteratively would that really make PICO an ineffective tool?  If some of these assumption were lifted and the resulting nonlinear model conserved mass and energy would that not be considered worthwhile?  More discussion about why these assumptions were made and their consequences would make the manuscript stronger and allow people to make more informed choices about adopting the PICO model in the future.

Minor comments:

Abstract Line 11.  Change "The two-dimensional melt rate fields …" with "We identify a set of parameters that yields two-dimensional melt rate fields that qualitatively reproduce the typical pattern of … "

Line 28.   I take issue with the use of the term "resolving" here because to me it implies that the PICO is driven with equations of fluid motion at a resolution fine enough to capture the overturning.  PICO parameterizes the ice shelf cavity transports associated with an imposed overturning circulation driven by the ice pump.

Page 5 line 15.  Of the many simplifying assumptions made here, why are the gamma T and gamma S parameters set as constant?  I don't see why you couldn't pull out an some kind of velocity in the PICO grid cells.  Just stating that you are following OH10 does not explain why the choice is made.  Would introducing velocity dependence interfere with the analytical solvability?

Page 6 line 5:  Neglecting heat fluxes into ice shelves is another odd choice.  As ice shelves thin or under warm ice shelves the conductive heat flux into ice can be about 10% of heat flux that melts ice.

Page 6 line 9-14 and E8: Just to be clear, you are solving for the melt rate using far field T and S (A5) and then using that melt rate to solve for the boundary layer T and S (A2 and A3)?

Page 6, line 10: The discussion in Holland and Jenkins 1999 describes the conditions in which simplified forms of the  3-equations model can be expected to yield similar results.  Is the PICO

model subjected to that same range of conditions or are there circumstances under which one might expect that your simplified equations might be expected to substantically deviate from the three equation solution?

Page 9, "exemplary shown"?

Page 10, Line 1-2:  This section describes a parameter tuning exercise, not a model validation exercise.  You are seeking a range of acceptable parameters using four criteria as constraints.

Page 12 Line 14: Provide the parameter values shown in OH10 and Holland and Jenkins 1999 here so the reader doesn't have to go digging.

Page 14, Line 6:  Change "Due to this model assumptions" to "Due to these model assumptions"

Page 14, Line 12: How large are the deviations of the Filchner-Ronne and Ross Ice shelves then?

Page 16, Line 24: Change to, "PICO does not resolve ocean dynamics.  PICO parameterizes vertical ocean circulation in the ice shelf cavities."   Later you say they do not "resolve horizontal ocean circulation"  Probably better to say "As PICO is a 2D box model, no horizontal flow variations are represented."

Page 17, Line 2-3.  I do not understand the sentence that begins, "A necessary condition for the box model to work..."

---

## Author Response (AR2)

**Response to Reviewer Comments**
**By R. Reese, T. Albrecht, M. Mengel, X. Asay-Davis and R. Winkelmann**

Journal: TC
Title: Antarctic sub-shelf melt rates via PICO
Author(s): R. Reese, T. Albrecht, M. Mengel, X. Asay-Davis and R. Winkelmann
MS No.: tc-2017-70
MS Type: Research article

We are very happy that our manuscript was accepted for publication in The Cryosphere. We would like to thank the editor Eric Larour, the reviewer Frank Pattyn and the second, anonymous reviewer for their helpful and excellent comments and their efforts to create the detailed reviews!

In the current manuscript version, we addressed the comments by the first, anonymous reviewer for the revised manuscript:

- We included the comments on 'Abstract, Line 11', 'Line 28', 'Page 9', 'Page 10, Line 1-2', 'Page 12, Line 14', 'Page 14, Line 6', 'Page 14, Line 12', 'Page 16, Line 24', 'Page 17, Line 2-3'. The changes made to the manuscript are highlighted in the latex-diff attached.

- We appreciate the major comments as well as the minor comments on 'Page 5, Line 15' and 'Page 6, Line 5' and plan to include them in an improved model version in the future.

- 'Page 6, line 9-14': We solve simultaneously for melt rates (Eq. 8), overturning $q$ (Eq. 3) as well as salinity and temperatures in the adjacent ocean box $T_k$ and $S_k$ (Eq. A12, A8 and A13, A8) and then diagnose boundary layer $T$ and $S$ (Eqs. A2 and A3).

- 'Page 6, line 10': We checked that the PICO model is subject to conditions under which the simplified form of the 3-equations model is expected to yield similar results.

The changes made to the main document can be found in the latex-diff document on the following pages.

Kind regards,

Ronja Reese on behalf of the co-authors.

[revised manuscript text omitted]